# Hypoxia-inducible factor induces cysteine dioxygenase and promotes cysteine homeostasis in *Caenorhabditis elegans*

Kurt Warnhoff[1,2]*, Sushila Bhattacharya[1], Jennifer Snoozy[1], Peter C Breen[3], Gary Ruvkun[3]

[1]Pediatrics and Rare Diseases Group, Sanford Research, Sioux Falls, United States; [2]Department of Pediatrics, Sanford School of Medicine, University of South Dakota, Sioux Falls, United States; [3]Department of Molecular Biology, Massachusetts General Hospital, Boston, United States

*For correspondence: kurt.warnhoff@sanfordhealth.org

Competing interest: The authors declare that no competing interests exist.

**Abstract** Dedicated genetic pathways regulate cysteine homeostasis. For example, high levels of cysteine activate cysteine dioxygenase, a key enzyme in cysteine catabolism in most animal and many fungal species. The mechanism by which cysteine dioxygenase is regulated is largely unknown. In an unbiased genetic screen for mutations that activate cysteine dioxygenase (*cdo-1*) in the nematode *Caenorhabditis elegans,* we isolated loss-of-function mutations in *rhy-1* and *egl-9,* which encode proteins that negatively regulate the stability or activity of the oxygen-sensing hypoxia inducible transcription factor (*hif-1*). EGL-9 and HIF-1 are core members of the conserved eukaryotic hypoxia response. However, we demonstrate that the mechanism of HIF-1-mediated induction of *cdo-1* is largely independent of EGL-9 prolyl hydroxylase activity and the von Hippel-Lindau E3 ubiquitin ligase, the classical hypoxia signaling pathway components. We demonstrate that *C. elegans cdo-1* is transcriptionally activated by high levels of cysteine and *hif-1*. *hif-1*-dependent activation of *cdo-1* occurs downstream of an $H_2S$-sensing pathway that includes *rhy-1*, *cysl-1*, and *egl-9*. *cdo-1* transcription is primarily activated in the hypodermis where it is also sufficient to drive sulfur amino acid metabolism. Thus, the regulation of *cdo-1* by *hif-1* reveals a negative feedback loop that maintains cysteine homeostasis. High levels of cysteine stimulate the production of an $H_2S$ signal. $H_2S$ then acts through the *rhy-1/cysl-1/egl-9* signaling pathway to increase HIF-1-mediated transcription of *cdo-1*, promoting degradation of cysteine via CDO-1.

## eLife assessment

The study presents **valuable** findings on how the hypoxia response pathway senses and responds to changes in the homeostasis of the amino acid cysteine and other sulfur-containing molecules. By providing a **compelling**, rigorous genetic analysis of the pathway, the study adds to a growing body of literature showing that prolyl hydroxylation is not the only mechanism by which the hypoxia response pathway can act. Although the paper does not reveal new biochemical insight into the mechanism, it opens up new areas of investigation that will be of interest to cell biologists and biomedical researchers studying the many pathologies involving hypoxia and/or cysteine metabolism.

## Introduction

Cysteine is a sulfur-containing amino acid that mediates many oxidation/reduction reactions of proteins, is the redox center of the abundant antioxidant tripeptide glutathione which also serves

**eLife digest** Proteins are large molecules in our cells that perform various roles, from acting as channels through which nutrients can enter the cell, to forming structural assemblies that help the cell keep its shape. Proteins are formed of chains of building blocks called amino acids. There are 20 common amino acids, each with a different 'side chain' that confers it with specific features.

Cysteine is one of these 20 amino acids. Its side chain has a 'thiol' group, made up of a sulfur atom and a hydrogen atom. This thiol group is very reactive, and it is an essential building block of enzymes (proteins that speed up chemical reactions within the cell), structural proteins and signaling molecules. While cysteine is an essential amino acid for the cell to function, excess cysteine can be toxic. The concentration of cysteine in animal cells is tightly regulated by an enzyme called cysteine dioxygenase.

This enzyme is implicated in two rare conditions that affect metabolism, where the product of cysteine dioxygenase is a key driver of disease severity. Additionally, cysteine dioxygenase acts as a tumor suppressor gene, and its activity becomes blocked in diverse cancers. Understanding how cysteine dioxygenase is regulated may be important for research into these conditions.

While it has been shown that excess cysteine drives the production and activity of cysteine dioxygenase, how the cell detects high levels of cysteine remained unknown. Warnhoff et al. sought to resolve this question using the roundworm *Caenorhabditis elegans*. First, the scientists demonstrated that, like in mammals, high levels of cysteine drive the production of cysteine dioxygenase in *C. elegans*. Next, the researchers used an approach called an unbiased genetic screening to find genes that induce cysteine dioxygenase production when they are mutated. These experiments revealed that the protein HIF-1 can drive the production of cysteine dioxygenase when it is activated by a pathway that senses hydrogen sulfide gas.

Based on these results, Warnhoff et al. propose that high levels of cysteine lead to the production of hydrogen sulfide gas that in turn drives the production of cysteine dioxygenase via HIF-1 activation of gene expression.

The results reported by Warnhoff et al. suggest that modulating HIF-1 signaling could control the activity of cysteine dioxygenase. This information could be used in the future to develop therapies for molybdenum cofactor deficiency, isolated sulfite oxidase deficiency and several types of cancer. However, first it will be necessary to demonstrate that the same signaling pathway is active in humans.

as a major cysteine reserve, and is essential for iron-sulfur cluster assembly in the mitochondrion (*Wu et al., 2004*; *Zheng et al., 1993*). Cysteine residues in many proteins are in close proximity in the primary or folded protein sequence and are oxidized in the endoplasmic reticulum to form intra- and interprotein disulfide linkages, most commonly in secreted proteins which mediate intercellular signaling and defense (*Noiva, 1994*; *Raina and Missiakas, 1997*; *Rietsch and Beckwith, 1998*). In many enzymes, the reactivity of the cysteine sulfur is key for the coordination of metals such as zinc or iron, which support protein structure and catalytic activity (*Giles et al., 2003*; *Tainer et al., 1991*; *Miller et al., 1985*). Cysteine is also a key source of hydrogen sulfide ($H_2S$), a volatile signaling molecule (*Singh and Banerjee, 2011*). While cysteine has these essential functions, excess cysteine is also toxic. High levels of cysteine impair mitochondrial respiration by disrupting iron homeostasis (*Hughes et al., 2020*), acts as a neural excitotoxin (*Olney et al., 1990*), and promotes the formation of toxic levels of hydrogen sulfide gas (*Singh and Banerjee, 2011*; *Evans, 1967*; *Truong et al., 2006*). Given this balance between essential and toxic, cysteine homeostasis is key for the health of cells and organisms.

CDO1-mediated oxidation is the primary pathway of cysteine catabolism when sulfur amino acid (methionine or cysteine) availability is normal or high (*Bella et al., 1996*). The dipeptide cystathionine is a key intermediate in this pathway. Cystathionine is catabolized by cystathionase (CTH-2 in *Caenorhabditis elegans*, CTH in mammals) producing cysteine and α-ketobutyrate. Cysteine is further oxidized using dissolved atmospheric dioxygen to cysteinesulfinate by cysteine dioxygenase (CDO-1 in *C. elegans*, CDO1 in mammals; *Figure 1A*; *Stipanuk, 2004*). The further oxidation of cysteinesulfinate downstream of CDO-1 generates highly toxic sulfites that are normally oxidized to more benign sulfate by sulfite oxidase (*Figure 1A*).

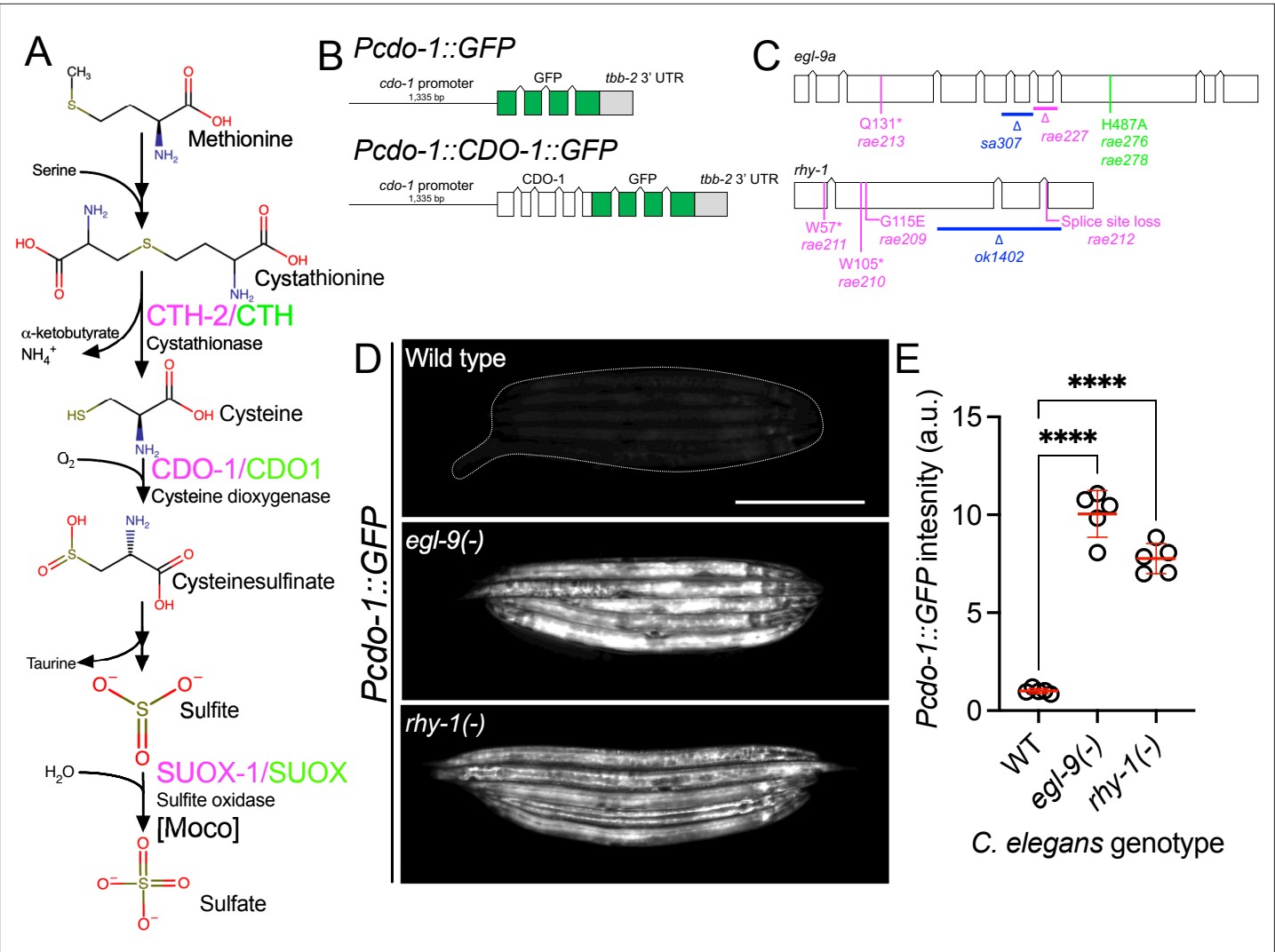

**Figure 1.** *egl-9* and *rhy-1* inhibit *cdo-1* transcription. (**A**) Pathway for sulfur amino acid metabolism beginning with methionine. We highlight the roles of cystathionase (CTH-2/CTH), cysteine dioxygenase (CDO-1/CDO1), and the Moco-requiring sulfite oxidase enzyme (SUOX-1/SUOX). *C. elegans* enzymes (magenta) and their human homologs (green) are displayed. (**B**) *Pcdo-1::GFP* promoter fusion (upper) and *Pcdo-1::CDO-1::GFP* C-terminal protein fusion (lower) transgenes used in this work are displayed. Boxes indicate exons, connecting lines indicate introns. The *cdo-1* promoter is shown as a straight line. (**C**) *egl-9a* and *rhy-1* gene structures. Boxes indicate exons and connecting lines are introns. Colored annotations indicate mutations generated or used in our work. Magenta; chemically-induced mutations that activated *Pcdo-1::CDO-1::GFP* fusion protein. Blue; reference null alleles isolated independent of our work. Green; CRISPR/Cas9-generated mutation that inactivates the prolyl hydroxylase domain of EGL-9. (**D**) Expression of *Pcdo-1::GFP* transgene is displayed for wild-type, *egl-9(sa307)*, and *rhy-1(ok1402) C. elegans* animals at the L4 stage. Scale bar is 250 µm. White dotted line outlines animals with basal GFP expression. For GFP imaging, exposure time was 100ms. (**E**) Quantification of GFP expression displayed in (**D**). Individual datapoints are shown (circles) as are the mean and standard deviation (red lines). *n* is 5 individuals per genotype. Data are normalized so that wild-type expression of *Pcdo-1::GFP* is 1 arbitrary unit (a.u.). ****, p<0.0001, ordinary one-way ANOVA with Dunnett's post hoc analysis.

The online version of this article includes the following figure supplement(s) for figure 1:

**Figure supplement 1.** The *Pcdo-1::CDO-1::GFP* transgene encodes a functional cysteine dioxygenase enzyme.

As a critical player in cysteine homeostasis, CDO1 is a highly regulated enzyme: the activity and abundance of CDO1 increase dramatically in cells and animals fed excess cysteine and methionine (**Stipanuk and Ueki, 2011**). CDO1 levels are governed both transcriptionally and post-translationally via proteasomal degradation (**Dominy et al., 2006a**; **Dominy et al., 2006b**; **Stipanuk et al., 2004**; **Lee et al., 2004**; **Kwon and Stipanuk, 2001**). However, the molecular players that sense high levels of cysteine and promote CDO1 activation remain incompletely defined.

Understanding these regulatory mechanisms is an important goal as CDO1 dysfunction is implicated in disease. CDO1 is a tumor suppressor whose activity is silenced in diverse cancers via promoter methylation, a potential biomarker of tumor grade and progression (*Kojima et al., 2018*; *Yamashita et al., 2018*; *Brait et al., 2012*). Decreased CDO1 activity may support tumor cell growth by reducing reactive oxygen species and decreasing drug susceptibility (*Kang et al., 2019*; *Jeschke et al., 2013*; *Hao et al., 2017*; *Ma et al., 2022*).

Using the nematode *C. elegans,* we have previously shown that CDO-1 is a key player in the pathophysiology of two fatal inborn errors of metabolism; isolated sulfite oxidase deficiency (ISOD) and molybdenum cofactor deficiency (MoCD; *Duran et al., 1978*; *Mudd et al., 1967*). Molybdenum cofactor (Moco) is an essential 520 Dalton prosthetic group synthesized from GTP by a conserved multistep biosynthetic pathway that is present in about 2/3 of bacterial genomes and nearly all eukaryotic genomes (*Schwarz et al., 2009*; *Zhang and Gladyshev, 2008*). *C. elegans* can either retrieve Moco synthesized by the bacteria it consumes or can synthesize Moco de novo using its own Moco biosynthetic pathway (*Warnhoff and Ruvkun, 2019*). *C. elegans* strains carrying mutations in genes encoding Moco biosynthetic enzymes (*moc*) and feeding on wild-type *E. coli* develop normally. Yet, these same *moc*-mutant *C. elegans* when fed on Moco-deficient *E. coli* as their sole nutritional source are inviable, arresting development at an early larval stage. A saturated genetic selection for mutations that suppress this inviability identified multiple independent mutations in *cth-2* or *cdo-1*, genes which encode enzymes in the cysteine biosynthetic and degradation pathway. The toxic sulfites produced downstream of CDO-1 are normally oxidized to more benign sulfate by Moco-requiring sulfite oxidase (SUOX-1 in *C. elegans,* SUOX in mammals) an essential enzyme in *C. elegans* and humans (*Figure 1A*; *Mudd et al., 1967*; *Warnhoff and Ruvkun, 2019*). Therefore, loss of *cdo-1* or *cth-2* suppresses the lethality caused by both Moco and sulfite oxidase deficiencies in *C. elegans* by preventing the production of toxic sulfites (*Warnhoff and Ruvkun, 2019*; *Warnhoff et al., 2021*; *Snoozy et al., 2022*). Thus, understanding the fundamental mechanisms that govern the levels and activity of CDO-1 is critical to generating new therapeutic hypotheses to treat these diseases.

To define genes that regulate *cdo-1* levels and activity, we performed an unbiased genetic screen in the nematode *C. elegans* to identify mutations that increase the expression or abundance of a P*cdo-1::CDO-1::GFP* reporter transgene. We identified multiple independent loss-of-function mutations in two genes, *egl-9* and *rhy-1*, that dramatically increase expression of this P*cdo-1::CDO-1::GFP* transgene. These enzymes act in the hypoxia and H$_2$S-sensing pathway, and we demonstrate that the conserved hypoxia-inducible transcription factor (HIF-1) activates *cdo-1* transcription in this pathway (*Shen et al., 2006*; *Budde and Roth, 2011*; *Ma et al., 2012*). We further show that high levels of cysteine promote *cdo-1* transcription, and that *hif-1* and *cysl-1* (another component of the H$_2$S-sensing pathway) are required for viability under high cysteine conditions. We demonstrate that transcriptional activation of *cdo-1* via HIF-1 promotes CDO-1 activity and establish the *C. elegans* hypodermis as a key tissue of CDO-1 activation and function. Unexpectedly, we find that *cdo-1* regulation is governed by a HIF-1 pathway largely independent of EGL-9 prolyl hydroxylase activity and von Hippel-Lindau (VHL-1), the canonical O$_2$ -sensing pathway (*Epstein et al., 2001*; *Ivan et al., 2001*). These data establish a new connection between the HIF-1/H$_2$S-sensing pathway and sulfur amino acid catabolism governed by CDO-1.

## Results

### *egl-9* and *rhy-1* negatively regulate *cdo-1* transcription

To identify regulators of CDO-1 expression or activity, we engineered a transgene expressing a C-terminal green fluorescent protein (GFP) fusion to the full-length CDO-1 protein driven by the native *cdo-1* promoter (P*cdo-1::CDO-1::GFP*, *Figure 1B*). Transgenic animals were generated by integrating the P*cdo-1::CDO-1::GFP* fusion protein into the *C. elegans* genome (*Frøkjær-Jensen et al., 2014*). The P*cdo-1::CDO-1::GFP* fusion protein was functional and rescued a *cdo-1* loss-of-function mutation: the P*cdo-1::CDO-1::GFP* fusion protein reverses the suppression of Moco-deficient lethality caused by *cdo-1* loss of function (*Figure 1—figure supplement 1*). The reanimation of Moco-deficient lethality by the transgene depends on CDO-1 enzymatic activity because a transgene expressing an active-site mutant P*cdo-1::CDO-1[C85Y]::GFP* does not rescue the *cdo-1* mutant phenotype (*Figure 1—figure supplement 1*; *Warnhoff and Ruvkun, 2019*; *McCoy et al., 2006*). Thus, the P*cdo-1::CDO-1::GFP*

transgenic fusion protein is functional, suggesting that its expression pattern reflects endogenous protein expression, localization, and levels.

We performed a mutagenesis screen to identify genes that control the expression or accumulation of CDO-1 protein. Specifically, we performed an EMS chemical mutagenesis of *C. elegans* and screened in the F2 generation, after newly induced random mutations were allowed to become homozygous, for mutations that caused increased GFP accumulation by the *Pcdo-1::CDO-1::GFP* reporter transgene (**Brenner, 1974**). Using whole-genome sequencing, we determined that two independently isolated mutants carried distinct mutations in *egl-9* and four other independently isolated mutants had unique mutations in *rhy-1* (**Figure 1C**). The presence of multiple independent alleles suggests these mutations in *egl-9* or *rhy-1* are causative for the increased *Pcdo-1::CDO-1::GFP* expression or accumulation observed. *egl-9* encodes the $O_2$-sensing prolyl hydroxylase and orthologue of mammalian EglN1. *rhy-1* encodes the regulator of hypoxia inducible transcription factor, an enzyme with homology to membrane-bound O-acyltransferases (**Shen et al., 2006**; **Epstein et al., 2001**). The genetic screen produced nonsense alleles of both *egl-9* and *rhy-1* suggesting the increased *Pcdo-1::CDO-1::GFP* expression or accumulation is caused by loss of *egl-9* or *rhy-1* function (**Figure 1C**). Inactivation of *egl-9* or *rhy-1* activates a transcriptional program mediated by the hypoxia inducible transcription factor, HIF-1 (**Shen et al., 2006**; **Epstein et al., 2001**). To determine if *egl-9* or *rhy-1* regulate *cdo-1* transcription, we engineered a separate reporter construct where only GFP (rather than the full length CDO-1::GFP fusion protein) is transcribed by the *cdo-1* promoter (*Pcdo-1::GFP*, **Figure 1B**). This *Pcdo-1::GFP* transgene was introduced into strains with independently isolated *egl-9(sa307)* and *rhy-1(ok1402)* null reference alleles (**Shen et al., 2006**; **Darby et al., 1999**). The *egl-9(sa307)* and *rhy-1(ok1402)* mutations dramatically induce expression of GFP driven by the *Pcdo-1::GFP* transcriptional reporter transgene (**Figure 1D and E**). The activation of *cdo-1* transcription by independently isolated *egl-9* or *rhy-1* mutations demonstrates that the *egl-9* and *rhy-1* mutations isolated in our screen for the induction or accumulation of *Pcdo-1::CDO-1::GFP* are the causative genetic lesions. Furthermore, these data demonstrate that *egl-9* and *rhy-1* are necessary for the normal transcriptional repression of *cdo-1*.

## HIF-1 directly activates *cdo-1* transcription downstream of the *rhy-1*, *cysl-1*, *egl-9* genetic pathway

*rhy-1* and *egl-9* act in a pathway that regulates the abundance and activity of the HIF-1 transcription factor (**Shen et al., 2006**; **Ma et al., 2012**). The activity of *rhy-1* is most upstream in the pathway and negatively regulates the activity of *cysl-1*, which encodes a cysteine synthase-like enzyme of probable algal origin (**Wang et al., 2022**). CYSL-1 directly binds to and inhibits EGL-9 in an $H_2S$-modulated manner (**Ma et al., 2012**). EGL-9 uses molecular oxygen as well as an α-ketoglutarate cofactor to directly inhibit HIF-1 via prolyl hydroxylation, which recruits the VHL-1 ubiquitin ligase to ubiquitinate HIF-1, targeting it for degradation by the proteasome (Figure 5A) (**Salceda and Caro, 1997**; **Huang et al., 1998**; **Sutter et al., 2000**). Given that loss-of-function mutations in *rhy-1* or *egl-9* activate *cdo-1* transcription, we tested if *cdo-1* transcription is activated by HIF-1 as an output of this hypoxia/$H_2S$-sensing pathway. We performed epistasis studies using null mutations that inhibit the activity of *hif-1* (*cysl-1(ok762)* and *hif-1(ia4)*) or activate *hif-1* (*rhy-1(ok1402)* and *egl-9(sa307)*). The induction of *Pcdo-1::GFP* by *egl-9* inactivation was dependent upon the activity of *hif-1*, but not on the activity of *cysl-1* (**Figure 2A and B**). In contrast, induction of *Pcdo-1::GFP* by *rhy-1* inactivation was dependent upon the activity of both *hif-1* and *cysl-1* (**Figure 2A and C**). These results reveal a genetic pathway whereby *rhy-1*, *cysl-1*, and *egl-9* function in a negative-regulatory cascade to control the activity of HIF-1 which transcriptionally activates *cdo-1*. Our epistasis studies of *cdo-1* transcriptional regulation by HIF-1 align well with previous analyses of this genetic pathway in the context of transcription of *cysl-2* (a paralog of *cysl-1*) and the 'O₂-ON response' (**Ma et al., 2012**).

To demonstrate that HIF-1 activates transcription of endogenous *cdo-1*, we explored published RNA-sequencing data of wild-type, *egl-9(-)*, and *egl-9(-) hif-1(-)* mutant animals (**Pender and Horvitz, 2018**). *egl-9(-)* mutant *C. elegans* display an eightfold increase in *cdo-1* mRNA compared to wild type. This induction was dependent on *hif-1*; a *hif-1(-)* mutation completely suppressed the induction of *cdo-1* mRNA caused by an *egl-9(-)* mutation (**Pender and Horvitz, 2018**). These RNA-seq data confirm our findings using the *Pcdo-1::GFP* transcriptional reporter that HIF-1 is a transcriptional activator of *cdo-1*. ChIP-seq data of HIF-1 performed by the modERN project show that HIF-1 directly

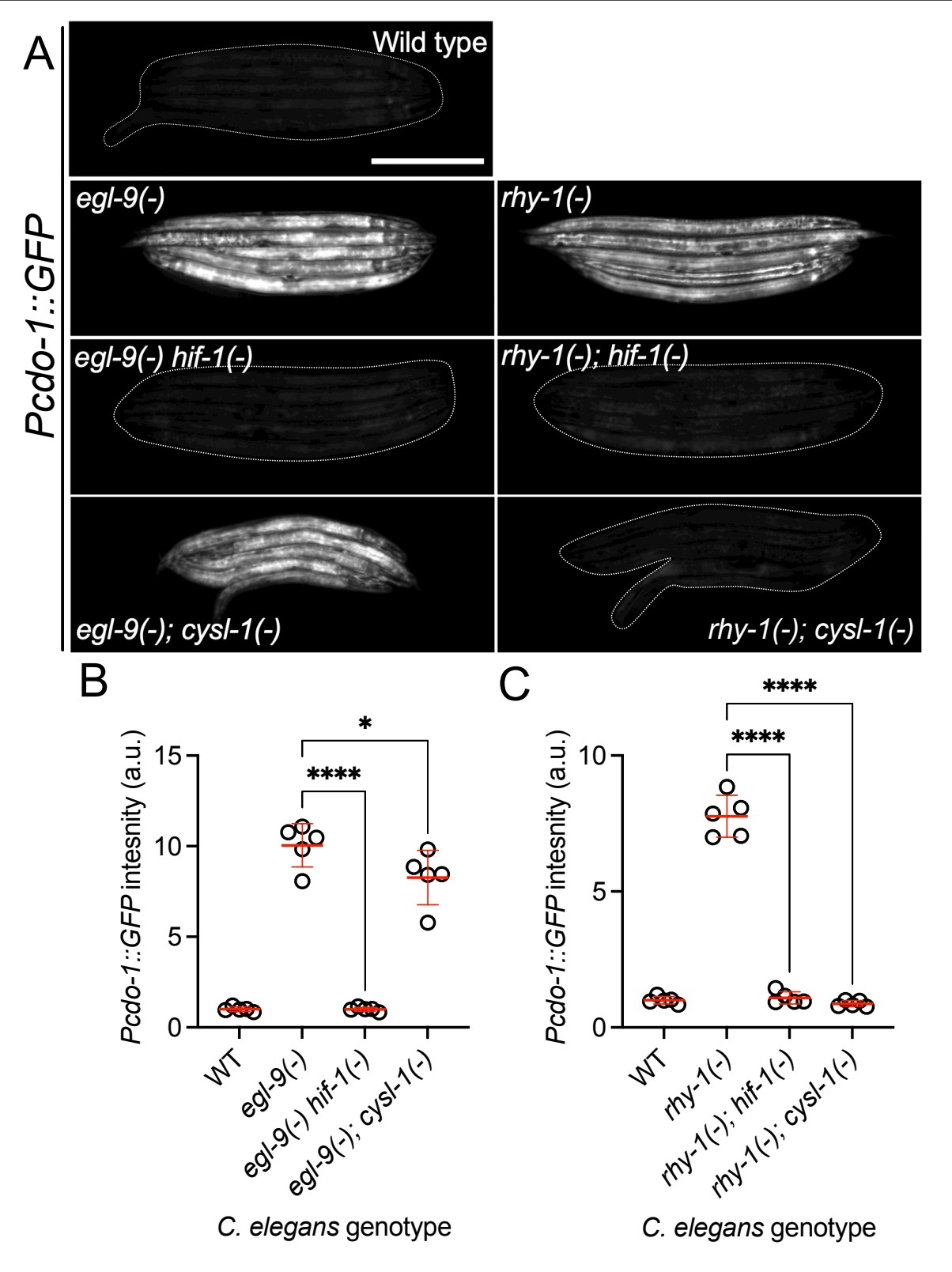

**Figure 2.** *cdo-1* transcription is activated by HIF-1 downstream of RHY-1, CYSL-1, and EGL-9. (**A**) Expression of the *Pcdo-1::GFP* transgene is displayed for wild-type, *egl-9(sa307)*, *egl-9(sa307) hif-1(ia4)* double mutant, *egl-9(sa307); cysl-1(ok762)* double mutant, *rhy-1(ok1402)*, *rhy-1(ok1402); hif-1(ia4)* double mutant, and *rhy-1(ok1402); cysl-1(ok762)* double mutant *C. elegans* animals at the L4 stage of development. Scale bar is 250 µm. White dotted line outlines animals with basal GFP expression. For GFP imaging, exposure time was 100ms. (**B, C**) Quantification of the data displayed in (**A**). Individual

*Figure 2 continued on next page*

*Figure 2 continued*

datapoints are shown (circles) as are the mean and standard deviation (red lines). *n* is 5 individuals per genotype. Data are normalized so that wild-type expression of *Pcdo-1::GFP* is 1 arbitrary unit (a.u.). *, p<0.05, ****, p<0.0001, ordinary one-way ANOVA with Dunnett's post hoc analysis. Note, wild-type, *egl-9(-)*, and *rhy-1(-)* images in panel A and quantification of *Pcdo-1::GFP* in panels B and C are identical to the data presented in (*Figure 1D and E*). They are re-displayed here to allow for clear comparisons to the double mutant strains of interest.

binds the *cdo-1* promoter (peak from −1165 to −714 base pairs 5′ to the *cdo-1* ATG start codon; *Kudron et al., 2018*; *Vora et al., 2022*). This HIF-1 binding site contains three copies of the HIF-binding motif (5′-RCGTG-3′; *Kaelin and Ratcliffe, 2008*). Thus, *cdo-1* is a downstream effector of HIF-1 and is likely a direct transcriptional target of HIF-1.

## High levels of cysteine promote *cdo-1* transcription and cause lethality in *cysl-1* and *hif-1* mutant animals

Mammalian CDO1 levels and activity are highly induced by dietary cysteine (*Bella et al., 1996*; *Stipanuk, 2004*; *Stipanuk and Ueki, 2011*). To determine if this homeostatic response is conserved in *C. elegans,* we exposed transgenic *C. elegans* carrying the *Pcdo-1::GFP* transcriptional reporter to high supplemental cysteine. Like our *egl-9* and *rhy-1* loss-of-function mutations, high levels of cysteine promoted *cdo-1* transcription (*Figure 3A–C*). Despite the significant 3.6-fold induction of *Pcdo-1::GFP* caused by 100 µM supplemental cysteine, we note that this induction is not as dramatic as the induction caused by null mutations in *egl-9* or *rhy-1*. This perhaps reflects the animals' ability to buffer environmental cysteine which is bypassed by genetic intervention.

We hypothesized that cysteine might activate *cdo-1* transcription through the RHY-1/CYSL-1/EGL-9/HIF-1 pathway. In this pathway, *cysl-1* and *hif-1* act to promote *cdo-1* transcription. Thus, we sought to test whether *cysl-1* or *hif-1* were necessary for the induction of *Pcdo-1::GFP* by high levels of cysteine. However, this experiment was not possible as we observed 100% lethality in *cysl-1(-)* and *hif-1(-)* mutant animals exposed to 100 µM supplemental cysteine, a cysteine concentration at which wild-type animals are healthy (*Figure 3E*). Importantly, wild-type, *cysl-1(-)* and *hif-1(-)* animals were all healthy under control conditions without supplemental cysteine (*Figure 3D*). While this phenotype limits our ability to establish the role of *cysl-1* and *hif-1* in the induction of *cdo-1* by high levels of cysteine, these data demonstrate that *cysl-1* and *hif-1* are necessary for survival under high cysteine conditions. Given the requirement of *hif-1* for survival in high levels of cysteine, we hypothesized that mutations in *egl-9* or *rhy-1* that activate *hif-1* might promote cysteine resistance. Indeed, we found that *egl-9(-)* and *rhy-1(-)* mutant *C. elegans* were partially viable when exposed to 1000 µM supplemental cysteine, a concentration that causes 100% lethality in wild-type animals (*Figure 3F*). Thus, *egl-9* and *rhy-1* are negative regulators cysteine tolerance. Taken together, these data demonstrate a critical physiological role for the RHY-1/CYSL-1/EGL-9/HIF-1 pathway in promoting cysteine homeostasis.

To further test the interaction between high levels of cysteine and the RHY-1/CYSL-1/EGL-9/HIF-1 pathway, we exposed *egl-9(-); Pcdo-1::GFP* mutant animals to control or high levels of supplemental cysteine. We reasoned that if cysteine and *egl-9* loss of function promote *Pcdo-1::GFP* accumulation in the same pathway, then their effects should not be additive. Consistent with this hypothesis, we saw no difference in *Pcdo-1::GFP* expression in *egl-9(-)* mutant animals exposed to 0 or 100 µM supplemental cysteine (*Figure 3A and B*). We also tested the ability of supplemental cysteine to further activate *Pcdo-1::GFP* expression in *rhy-1(-)* mutant animals. In contrast to our results with *egl-9(-)*, we observed that high levels of cysteine caused a significant induction of *Pcdo-1::GFP* in the *rhy-1(-)* mutant background (*Figure 3A and B*). These data suggest that cysteine acts in a pathway with *egl-9* but operates in parallel to the function of *rhy-1*.

Given the established role of CDO-1 in cysteine catabolism, we tested whether *cdo-1(-)* mutants were also sensitive to high levels of cysteine. *cdo-1(-)* mutant animals were not sensitive to high levels of supplemental cysteine compared to the wild type (*Figure 3D–F*). Thus, *cdo-1* is not necessary for survival under high cysteine conditions. This suggests the existence of alternate pathways that promote cysteine homeostasis. Given the role of the RHY-1/CYSL-1/EGL-9/HIF-1 pathway in promoting cysteine homeostasis, we propose that HIF-1 activates pathways (in addition to *cdo-1*) that promote survival under high cysteine conditions.

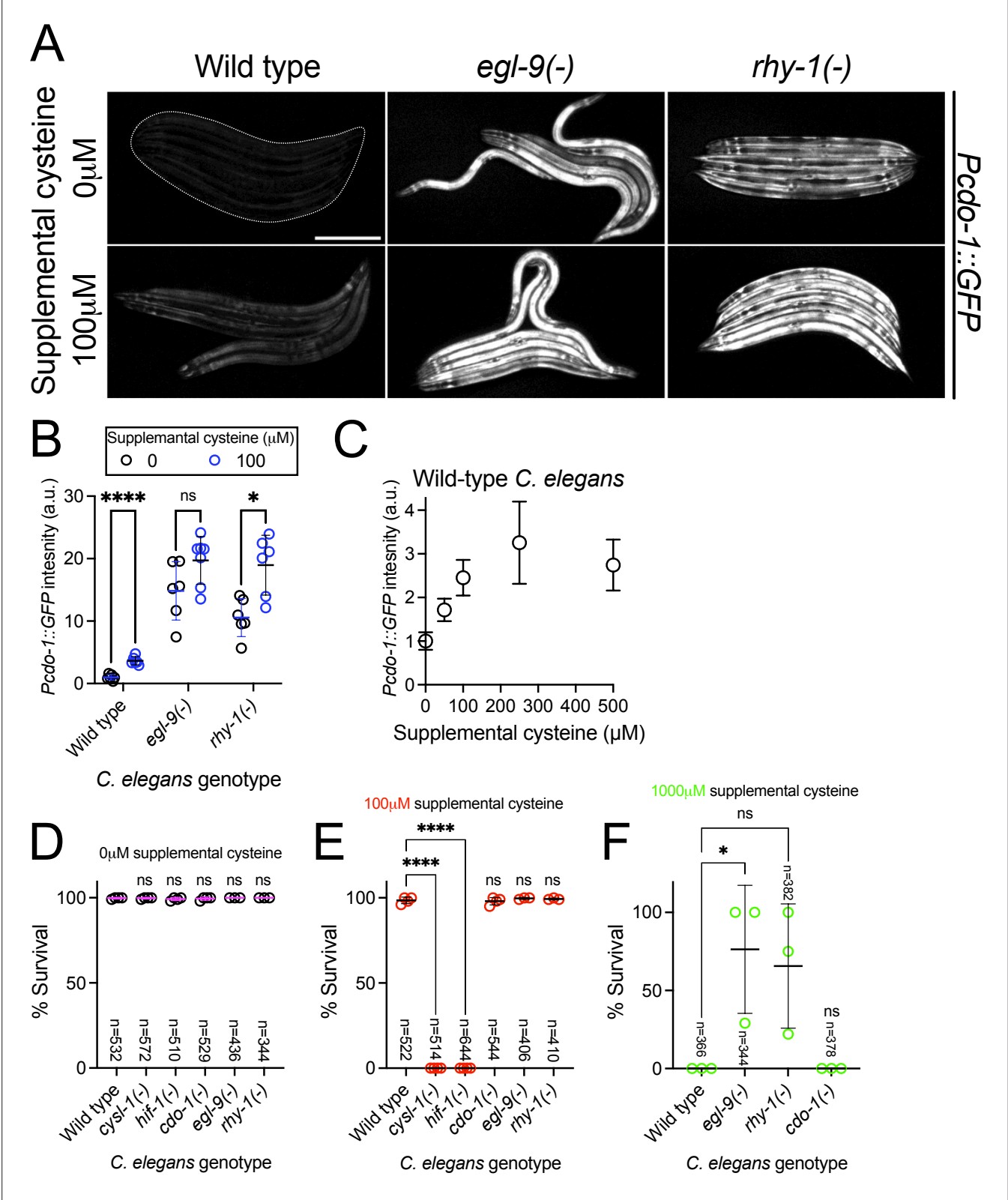

**Figure 3.** High levels of cysteine activate *cdo-1* transcription and are lethal to *hif-1* and *cysl-1* mutant animals. (**A**) Expression of the *Pcdo-1::GFP* transgene is displayed for wild-type, *egl-9(sa307),* and *rhy-1(ok1402)* young-adult *C. elegans* exposed to 0 or 100 μM supplemental cysteine. Scale bar is 250 μm. White dotted line outlines animals with basal GFP expression. For GFP imaging, exposure time was 100ms. Supplemental cysteine did not impact the mortality of the animals being imaged. (**B**) Quantification of the data displayed in (**A**). Individual datapoints are shown (circles) as are the

*Figure 3 continued on next page*

*Figure 3 continued*

mean and standard deviation (black lines). *n* is 6 or 7 individuals per genotype. Data are normalized so that expression of *Pcdo-1::GFP* in wild-type *C. elegans* exposed to 0 µM supplemental cysteine is equal to 1 arbitrary unit (a.u.). *, p<0.05, ****, p<0.0001, multiple unpaired t test with Welch's correction. (**C**) Quantification of the *Pcdo-1::GFP* expression is displayed for wild-type young-adult *C. elegans* exposed to 0, 50, 100, 250, or 500 µM supplemental cysteine. Mean and standard deviation are displayed. n is 6 or 7 individuals per concentration of supplemental cysteine. (**D–F**) The percentage of animals that survive overnight exposure to (**D**) 0, (**E**) 100, or (**F**) 1000 µM supplemental cysteine. Individual datapoints (circles) represent biological replicates. Three or four biological replicates were performed for each experiment and the total individuals scored amongst all replicates is displayed (n). *, p<0.05, ****, p<0.0001, ordinary one-way ANOVA with Dunnett's post hoc analysis. ns indicates no significant difference was identified.

## Activated CDO-1 accumulates and is functional in the hypodermis

We sought to identify the site of action of CDO-1. To observe CDO-1 localization, we used CRISPR/Cas9 to insert the GFP open-reading frame into the endogenous *cdo-1* locus, replacing the native *cdo-1* stop codon. This *cdo-1(rae273)* allele encodes a C-terminal tagged 'CDO-1::GFP' fusion protein from the native *cdo-1* genomic locus (*Figure 4A*). To determine if CDO-1::GFP was functional, we combined the CDO-1::GFP fusion protein with a null mutation in *moc-1*, a gene that is essential for *C. elegans* Moco biosynthesis (*Warnhoff and Ruvkun, 2019*). We then observed the growth of *moc-1(-)* CDO-1::GFP *C. elegans* on wild-type and Moco-deficient *E. coli*. The *moc-1(-)* mutant *C. elegans* expressing CDO-1::GFP from the native *cdo-1* locus arrest during larval development when fed Moco-deficient *E. coli,* but not when fed Moco-producing *E. coli* (*Figure 4C*). This lethality is caused by the CDO-1-mediated production of sulfites which are only toxic when *C. elegans* is Moco deficient, and demonstrates that the CDO-1::GFP fusion protein is functional (*Warnhoff and Ruvkun, 2019*). These data suggest that the CDO-1::GFP expression we observe is physiologically relevant.

When wild-type animals expressing CDO-1::GFP were grown under standard culture conditions, we observed CDO-1::GFP expression in multiple tissues, including prominent expression in the hypodermis (*Figure 4B*, *Figure 4—figure supplement 1*). We tested if CDO-1::GFP fusion protein levels were affected by inactivation of *egl-9* or *rhy-1*. We generated *egl-9(-)*; CDO-1::GFP and *rhy-1(-)*; CDO-1::GFP animals, and assayed expression of CDO-1::GFP. We found that CDO-1::GFP fusion protein levels, encoded by *cdo-1(rae273),* were increased by *egl-9(-)* or *rhy-1(-)* mutations (*Figure 4B*, *Figure 4—figure supplement 1*). This is consistent with our studies using *Pcdo-1::CDO-1::GFP* and *Pcdo-1::GFP* transgenes. Furthermore, high levels of cysteine also promoted accumulation of CDO-1::GFP (*Figure 4—figure supplement 2*). In all scenarios, the hypodermis was the most prominent site of CDO-1::GFP accumulation. Specifically, we found that CDO-1::GFP was expressed in the cytoplasm of Hyp7, the major *C. elegans* hypodermal cell (*Figure 4B*).

Based on the expression pattern of CDO-1::GFP encoded by *cdo-1(rae273)*, we hypothesized that CDO-1 acts in the hypodermis to promote sulfur amino acid metabolism. To test this hypothesis, we engineered a *cdo-1* rescue construct in which *cdo-1* is expressed exclusively in the hypodermis under the control of a hypoderm-specific collagen (*col-10*) promoter (*Pcol-10::CDO-1::GFP*; *Hong et al., 2000*). Collagens are expressed exclusively in the hypodermis of nematodes, with a periodic induction in phase with the almost diurnal molting cycle (*Meeuse et al., 2020*). Multiple independent transgenic *C. elegans* strains were generated by integrating the *Pcol-10::CDO-1::GFP* construct into the *C. elegans* genome and tested for rescue of the *cdo-1(-)* mutant suppression of Moco-deficient larval arrest (*Frøkjær-Jensen et al., 2014*). Thus, tissue-specific complementation of the *cdo-1* mutation would regenerate a lethal arrest phenotype caused by Moco deficiency. We found that multiple independent transgenic strains of *cdo-1(-) moc-1(-)* double mutant animals carrying the *Pcol-10::CDO-1::GFP* transgene displayed a larval arrest phenotype when fed a Moco-deficient diet (*Figure 4D*). These data demonstrated that hypodermal-specific expression of *cdo-1* is sufficient to rescue the *cdo-1(-)* mutant suppression of Moco-deficient lethality. This rescue was dependent upon the enzymatic activity of CDO-1 as an active site variant of this transgene (*Pcol-10::CDO-1[C85Y]::GFP*) did not rescue the suppressed larval arrest of *cdo-1(-) moc-1(-)* double mutant animals fed Moco-deficient diets (*Figure 4D*). Taken together, our analyses of CDO-1::GFP expression demonstrate that CDO-1 is expressed, and that expression is regulated, in multiple tissues, principal among them being the hypodermis and Hyp7 cell. Our tissue-specific rescue data demonstrate that hypodermal expression of *cdo-1* is sufficient to promote cysteine catabolism and suggest that the hypodermis is a critical tissue for sulfur amino acid metabolism. However, we cannot exclude the possibility that CDO-1 also acts in other cells and tissues as well.

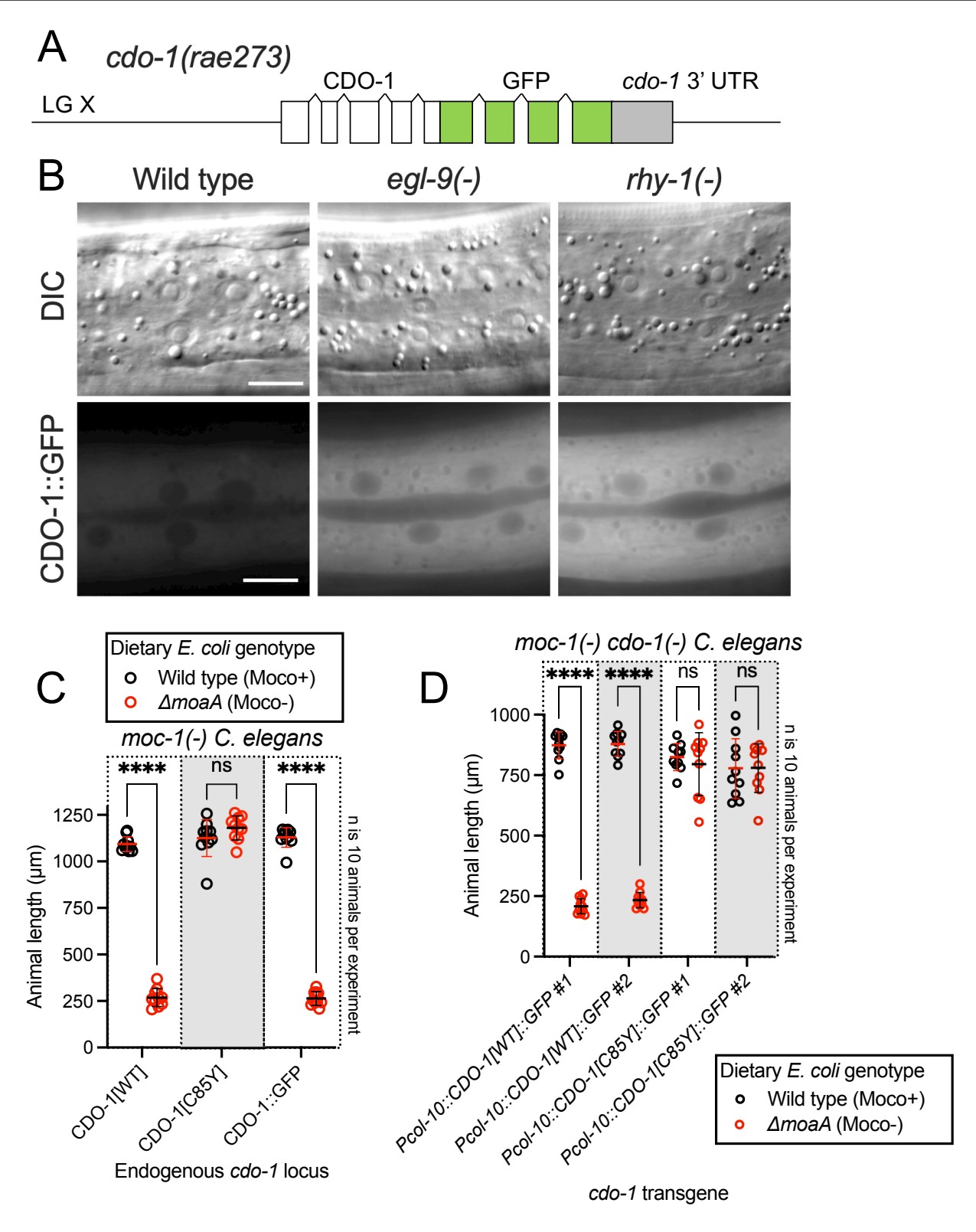

**Figure 4.** Hypodermal CDO-1 accumulates in the cytoplasm when *egl-9* or *rhy-1* are inactive and is sufficient to promote sulfur amino acid metabolism. (**A**) Diagram of *cdo-1(rae273)*, a CRISPR/Cas9-generated allele with GFP inserted into the *cdo-1* gene, creating a functional C-terminal CDO-1::GFP fusion protein expressed from the native *cdo-1* locus. (**B**) Differential interference contrast (DIC) and fluorescence imaging are shown for wild-type, *egl-9(sa307)*, and *rhy-1(ok1402) C. elegans* expressing CDO-1::GFP encoded by *cdo-1(rae273)*. Scale bar is 10 μm. For GFP imaging, exposure time was

*Figure 4 continued on next page*

Figure 4 continued

200 ms. An anterior segment of the Hyp7 hypodermal cell is displayed. CDO-1::GFP accumulates in the cytoplasm and is excluded from the nuclei. (**C**) *moc-1(ok366)*, *moc-1(ok366) cdo-1(mg622)*, and *moc-1(ok366) cdo-1(rae273)* animals were cultured from synchronized L1 larvae for 72 hr on wild-type (black, Moco+) or Δ*moaA* mutant (red, Moco-) *E. coli*. (**D**) *moc-1(ok366) cdo-1(mg622)* double mutant animals expressing *Pcol-10::CDO-1::GFP* or *Pcol-10::CDO-1[C85Y]::GFP* transgenes were cultured for 48 hr on wild-type (black, Moco+) or Δ*moaA* mutant (red, Moco-) *E. coli*. Two independently derived strains were tested for each transgene. For panels C and D, animal lengths were determined for each condition. Individual datapoints are shown (circles) as are the mean and standard deviation. Sample size (**n**) is 10 individuals for each experiment. \*\*\*\*, p<0.0001, multiple unpaired t test with Welch's correction. ns indicates no significant difference was identified.

The online version of this article includes the following figure supplement(s) for figure 4:

**Figure supplement 1.** A functional CDO-1::GFP fusion protein is induced by loss of *egl-9* or *rhy-1*.

**Figure supplement 2.** CDO-1::GFP encoded by *cdo-1(rae273)* is induced by supplemental cysteine.

## HIF-1 promotes CDO-1 activity downstream of the H₂S-sensing pathway

We sought to determine the physiological impact of *cdo-1* transcriptional activation by HIF-1. We reasoned mutations that activate HIF-1 and increase *cdo-1* transcription may cause increased CDO-1 activity. CDO-1 sits at a critical metabolic node in the degradation of the sulfur amino acids cysteine and methionine (*Figure 1A*). A key byproduct of sulfur amino acid metabolism and CDO-1 is sulfite, a reactive toxin that is detoxified by the Moco-requiring sulfite oxidase (SUOX-1). Null mutations in *suox-1* cause larval lethality. However, animals carrying the *suox-1(gk738847)* hypomorphic allele are healthy under standard culture conditions (*Warnhoff and Ruvkun, 2019*; *Warnhoff et al., 2021*). *suox-1(gk738847)* mutant animals display only 4% SUOX-1 activity compared to wild type and are exquisitely sensitive to sulfite stress (*Oliphant et al., 2023*). Thus, the *suox-1(gk738847)* mutation creates a sensitized genetic background to probe for increases in endogenous sulfite production. To test if increased *cdo-1* transcription would impact the growth of *suox-1*-comprimised animals, we combined the *egl-9* null mutation, which promotes HIF-1 activity and *cdo-1* transcription, with the *suox-1(gk738847)* allele. While *egl-9(-)* and *suox-1(gk738847)* single mutant animals are healthy under standard culture conditions, the *egl-9(-); suox-1(gk738847)* double mutant animals are extremely sick and require significantly more days to exhaust their *E. coli* food source under standard culture conditions (*Table 1*). These data establish a synthetic genetic interaction between these loci. To determine the role of sulfur amino acid metabolism in the *egl-9(-); suox-1(gk738847)* synthetic sickness phenotype, we engineered *egl-9(-); cdo-1(-) suox-1(gk738847)* and *cth-2(-); egl-9(-); suox-1(gk738847)* triple mutant animals. The *egl-9; suox-1* synthetic sickness phenotype was suppressed by inactivating mutations in *cdo-1* or *cth-2* which block the endogenous production of sulfite (*Table 1*). These data demonstrate that the deleterious activity of the *egl-9(-)* mutation in a *suox-1(gk738847)* background requires functional sulfur amino acid metabolism.

To determine the role of the H₂S-sensing pathway in the synthetic sickness phenotype displayed by *egl-9(-); suox-1(gk738847)* double mutant animals, we introduced null alleles of *cysl-1* or *hif-1* into the *egl-9(-); suox-1(gk738847)* double mutant. The *egl-9; suox-1* synthetic sickness phenotype was dependent upon *hif-1* but not *cysl-1* (*Table 1*). These results are consistent with our proposed genetic pathway and support the model that transcriptional activation of *cdo-1* by HIF-1 causes increased CDO-1 activity and increased flux of sulfur amino acids through their catabolic pathway.

Loss of *rhy-1* also strongly activates *cdo-1* transcription. We hypothesized that *rhy-1(-); suox-1(gk738847)* double mutant animals would display a synthetic sickness phenotype like *egl-9(-); suox-1(gk738847)* double mutant animals. However, based upon their ability to exhaust their *E. coli* food source under standard culture conditions, *rhy-1(-); suox-1(gk738847)* double mutant animals were just as healthy as either *rhy-1(-)* or *suox-1(gk738847)* single mutant *C. elegans* (*Table 1*). These data suggest that increasing *cdo-1* transcription alone is not sufficient to promote sulfite production via CDO-1. In addition to the role played by *rhy-1* in the regulation of HIF-1 activity, *rhy-1* itself is a transcriptional target of HIF-1. Loss of *egl-9* activity induces *rhy-1* mRNA ~50-fold in a *hif-1*-dependent manner (*Pender and Horvitz, 2018*). Given this potent transcriptional activation, we wondered if *rhy-1* might play an additional role downstream of HIF-1 in the regulation of sulfur amino acid metabolism and sulfite production. To test this hypothesis, we engineered *rhy-1(-); egl-9(-); suox-1(gk738847)* triple mutant animals. To our surprise, the *rhy-1(-); egl-9(-); suox-1(gk738847)* triple mutant animals

**Table 1.** Growth of *C. elegans* strains on standard laboratory conditions.

*C. elegans* strains and their corresponding mutations are displayed. For each strain, 5 L4-stage animals were seeded onto standard NGM petri dishes seeded with a monoculture of *E. coli* OP50. Petri dishes were monitored until the animals (and their progeny) depleted the lawn of *E. coli*. This was recorded as 'days for population to starve petri dish'. The average of these experiments is displayed for each *C. elegans* strain as is the standard deviation (SD) and the number of biological replicates (*n*). Significant differences in population growth were determined by appropriate comparisons to either *suox-1(gk738847)* (GR2269) or *egl-9(sa307); suox-1(gk738847)* (USD421) using an ordinary one-way ANOVA with Dunnett's post hoc analysis. No test indicates a statistical comparison was not made. Note, the data for USD421 are displayed twice in the table to allow for ease of comparison.

| *C. elegans* strain | Genetic locus 1 | Genetic locus 2 | Genetic locus 3 | Days for population to starve petri dish ±SD (n) | Significant difference compared to GR2269 (Adjusted p Value) |
|---|---|---|---|---|---|
| N2 | Wild type | Wild type | Wild type | 5±1 (8) | No test |
| GR2269 | Wild type | *suox-1(gk738847)* | Wild type | 8±1 (4) | No test |
| JT307 | *egl-9(sa307)* | Wild type | Wild type | 7±1 (5) | No test |
| USD421 | *egl-9(sa307)* | *suox-1(gk738847)* | Wild type | 19±5 (11) | 0.0003 (***) |
| USD926 | *egl-9(rae276)* | Wild type | Wild type | 6±1 (3) | No test |
| USD937 | *egl-9(rae276)* | *suox-1(gk738847)* | Wild type | 9±1 (3) | 0.93 (ns) |
| USD512 | *rhy-1(ok1402)* | Wild type | Wild type | 6±0 (4) | No test |
| USD414 | *rhy-1(ok1402)* | *suox-1(gk738847)* | Wild type | 7±0 (4) | 0.99 (ns) |
| CB5602 | *vhl-1(ok161)* | Wild type | Wild type | 6±0 (4) | No test |
| USD422 | *vhl-1(ok161)* | *suox-1(gk738847)* | Wild type | 8±1 (4) | 0.99 (ns) |

| *C. elegans* strain | Genetic locus 1 | Genetic locus 2 | Genetic locus 3 | Days for population to starve petri dish ±SD (n) | Significant difference compared to USD421 (Adjusted p value) |
|---|---|---|---|---|---|
| USD421 | *egl-9(sa307)* | *suox-1(gk738847)* | Wild type | 19±5 (11) | No test |
| USD430 | *egl-9(sa307)* | *suox-1(gk738847)* | *cdo-1(mg622)* | 7±0 (6) | <0.0001 (****) |
| USD433 | *egl-9(sa307)* | *suox-1(gk738847)* | *cth-2(mg599)* | 9±1 (4) | <0.0001 (****) |
| USD432 | *egl-9(sa307)* | *suox-1(gk738847)* | *rhy-1(ok1402)* | 7±1 (4) | <0.0001 (****) |
| USD434 | *egl-9(sa307)* | *suox-1(gk738847)* | *cysl-1(ok762)* | 16±1 (9) | 0.1 (ns) |
| USD431 | *egl-9(sa307)* | *suox-1(gk738847)* | *hif-1(ia4)* | 9±1 (6) | <0.0001 (****) |

were healthy, demonstrating that *rhy-1* was necessary for the deleterious activity of the *egl-9(-)* mutation in a *suox-1(gk738847)* background (**Table 1**). These genetic data suggest a dual role for *rhy-1* in the control of sulfur amino acid metabolism; first as a component of a regulatory cascade that controls the activity of HIF-1 and second as a functional downstream effector of HIF-1 that is required for sulfur amino acid metabolism.

This is not the first description of a *rhy-1* role downstream of *hif-1*. Overexpression of a *rhy-1*-encoding transgene suppresses the lethality of a *hif-1(-)* mutant during H$_S$S stress (**Horsman et al., 2019**). These data establish RHY-1 as both a regulator and effector of HIF-1. How RHY-1, a predicted membrane-bound O-acyltransferase, molecularly executes these dual roles remains to be explored.

## EGL-9 prolyl hydroxylase activity and VHL-1 are largely dispensable in the regulation of CDO-1

EGL-9 inhibits HIF-1 through its prolyl hydroxylase domain that hydroxylates HIF-1 proline 621 (**Figure 5A**; **Epstein et al., 2001**). To evaluate the impact of the EGL-9 prolyl hydroxylase domain on the regulation of *cdo-1*, we generated a prolyl hydroxylase domain-inactive *egl-9* mutation using CRISPR/Cas9. We engineered an *egl-9* mutation that substitutes an alanine in place of histidine 487 (H487A) (**Figure 1C**). Histidine 487 of EGL-9 is highly conserved and catalytically essential in the prolyl hydroxylase domain active site (**Figure 5B**; **Pan et al., 2007**; **Shao et al., 2009**). To evaluate the impact of an inactive EGL-9 prolyl hydroxylase domain on the transcription of *cdo-1*, we engineered a *C. elegans* strain carrying the *egl-9(H487A)* mutation with the *Pcdo-1::GFP* transcriptional reporter.

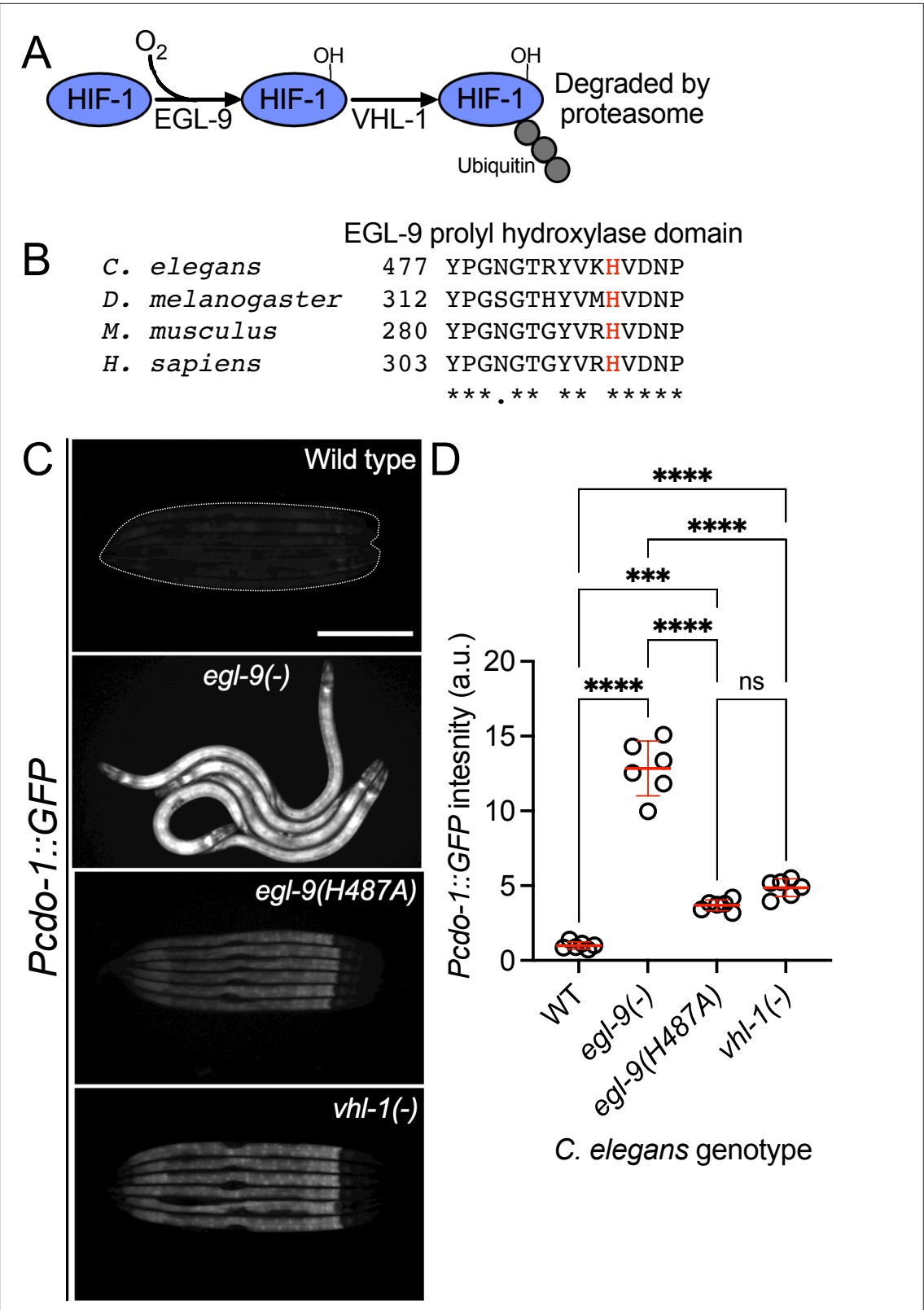

**Figure 5.** *egl-9* inhibits *cdo-1* transcription in a largely prolyl-hydroxylase and VHL-1-independent manner. (**A**) The pathway of HIF-1 processing during normoxia is displayed. EGL-9 uses $O_2$ as a substrate to hydroxylate (-OH) HIF-1 on specific proline residues. Prolyl hydroxylated HIF-1 is bound by VHL-1 which facilitates HIF-1 polyubiquitination and targets HIF-1 for degradation by the proteasome. (**B**) Amino acid alignment of the EGL-9 prolyl hydroxylase domain from *C. elegans, D. melanogaster, M. musculus,* and *H. sapiens.* '*' indicate perfect amino acid conservation while '.' indicates weak

*Figure 5 continued on next page*

*Figure 5 continued*

similarity amongst species compared. Highlighted (red) is the catalytically essential histidine 487 residue in *C. elegans*. Alignment was performed using Clustal Omega (EMBL-EBI). (**C**) Expression of *Pcdo-1::GFP* promoter fusion transgene is displayed for wild-type, *egl-9(sa307, -)*, *egl-9(rae276*, H487A), and *vhl-1(ok161) C. elegans* animals at the L4 stage of development. Scale bar is 250 µm. White dotted line outlines animals with basal GFP expression. For GFP imaging, exposure time was 500ms. (**D**) Quantification of the data displayed in (**C**). Individual datapoints are shown (circles) as are the mean and standard deviation (red lines). *n* is 6 individuals per genotype. Data are normalized so that wild-type expression of *Pcdo-1::GFP* is 1 arbitrary unit (a.u.). ***, $p < 0.001$, ****, $p < 0.0001$, ordinary one-way ANOVA with Tukey's multiple comparisons test. ns indicates no significant difference was identified.

*egl-9(H487A)* caused a modest increase in *Pcdo-1::GFP* accumulation in the *C. elegans* intestine, suggesting that the EGL-9 prolyl hydroxylase domain is necessary to repress *cdo-1* transcription in the intestine (*Figure 5C and D*). However, the activation of *Pcdo-1::GFP* by the *egl-9(H487A)* mutation was markedly less when compared to *Pcdo-1::GFP* activation caused by an *egl-9(-)* null mutation (*Figure 5C and D*). These data suggest that EGL-9 has a prolyl hydroxylase domain-independent activity that is responsible for repressing *cdo-1* transcription.

EGL-9 hydroxylates specific proline residues on HIF-1. These hydroxylated proline residues are recognized by the von Hippel-Lindau E3 ubiquitin ligase (VHL-1). VHL-1-mediated ubiquitination promotes degradation of HIF-1 by the proteasome (*Figure 5A*; *Ivan et al., 2001*). Thus, the EGL-9 prolyl hydroxylase domain and VHL-1 act in a pathway to regulate HIF-1. To determine the role of VHL-1 in regulating *cdo-1,* we engineered a *C. elegans* strain carrying the *vhl-1(-)* null mutation with our P*cdo-1::GFP* reporter. *vhl-1* inactivation also caused a modest increase in *Pcdo-1::GFP* expression in the *C. elegans* intestine, suggesting that *vhl-1* is necessary to repress *cdo-1* transcription (*Figure 5C and D*). However, activation of *Pcdo-1::GFP* by the *vhl-1(-)* mutation was less than activation caused by an *egl-9(-)* null mutation (*Figure 5C and D*). These data suggest that EGL-9 has a VHL-1-independent activity that is responsible for repressing *cdo-1* transcription.

To evaluate the impact of inactivating the EGL-9 prolyl hydroxylase domain or VHL-1 on cysteine metabolism, we again employed the *suox-1(gk738847)* hypomorphic mutation that sensitizes animals to increases in sulfite. We engineered *egl-9(H487A); suox-1(gk738847)* and *vhl-1(-) suox-1(gk738847)* double mutant animals and evaluated the health of those strains. In contrast to *egl-9(-); suox-1(gk738847)* double mutant animals which are extremely sick, *egl-9(H487A); suox-1(gk738847)* and *vhl-1(-) suox-1(gk738847)* double mutant animals are healthy (*Table 1*). These genetic data suggest that neither the EGL-9 prolyl hydroxylase domain nor VHL-1 are necessary to repress cysteine catabolism/sulfite production. However, it is plausible that the *egl-9(H487)* or *vhl-1(-)* mutations modestly activate cysteine metabolism, likely proportional to their activation of the *Pcdo-1::GFP* transgene (*Figure 5C and D*), and that this activation is not sufficient to produce enough sulfites to negatively impact the growth of *suox-1(gk738847)* mutant animals.

## Discussion
### CDO-1 is a physiologically relevant effector of HIF-1

The hypoxia-inducible factor HIF-1 is a master regulator of the cellular response to hypoxia. It activates the transcription of many genes and pathways that are critical to maintain metabolic homeostasis in the face of low $O_2$. For example, mammalian HIF1α induces the hematopoietic growth hormone erythropoietin, glucose transport and glycolysis, as well as lactate dehydrogenase (*Semenza and Wang, 1992*; *Bashan et al., 1992*; *Loike et al., 1992*; *Firth et al., 1994*). The nematode *C. elegans* encounters a range of $O_2$ tensions in its natural habitat of rotting material: as microbial abundance increases, $O_2$ levels decrease from atmospheric levels (~21% $O_2$). In fact, *C. elegans* prefers 5–12% $O_2$, perhaps because hypoxia predicts abundant bacterial food sources (*Gray et al., 2004*). Members of the HIF-1 pathway and its targets have emerged from genetic studies of *C. elegans* (*Epstein et al., 2001*). For instance, *C. elegans* HIF-1 promotes $H_2S$ homeostasis by inducing transcription of the mitochondrial sulfide quinone oxidoreductase (*sqrd-1),* detoxifying $H_2S$ (*Budde and Roth, 2011*).

We sought to define genes that regulate the levels and activity of cysteine dioxygenase (CDO-1), a critical regulator of cysteine homeostasis (*Bella et al., 1996*; *Stipanuk, 2004*; *Stipanuk and Ueki, 2011*). Taking an unbiased genetic approach in *C. elegans,* we found that *cdo-1* was highly regulated by HIF-1 downstream of a signaling pathway that includes *rhy-1*, *cysl-1*, and *egl-9*. We demonstrated that HIF-1 promotes *cdo-1* transcription, accumulation of CDO-1 protein, and increased CDO-1

activity. Loss of *rhy-1* or *egl-9* activate *hif-1* to in turn strongly induce the *cdo-1* promoter. The pathway for activation of *cdo-1* also requires *cysl-1*, which functions downstream of *rhy-1* and upstream of *egl-9*. Based on ChIP-Seq studies, HIF-1 directly binds to the *cdo-1* promoter (**Kudron et al., 2018**). Interestingly, mammalian CDO1 is regulated at both the transcriptional and post-transcriptional level in response to high dietary cysteine (**Dominy et al., 2006a**; **Dominy et al., 2006b**; **Stipanuk et al., 2004**; **Lee et al., 2004**; **Kwon and Stipanuk, 2001**). Our studies in the nematode *C. elegans* suggest that CDO1 transcription in mammals might be governed by HIF1α in response to changes in cellular cysteine. Importantly, all members of the RHY-1/CYSL-1/EGL-9 pathway in *C. elegans* have homologs encoded by mammalian genomes (**Ma et al., 2012**). Given the conservation of these proteins, future studies may show that similar cysteine and $H_2S$-responsive signaling pathways operate in mammals.

We also show that the transcriptional activation of *cdo-1* by HIF-1 promotes CDO-1 enzymatic activity. Genetic activation of *cdo-1* by loss of *egl-9* causes severe sickness in a mutant with reduced sulfite oxidase activity, an activity required to cope with the toxic sulfite produced via CDO-1. This synthetic sickness is dependent upon an intact HIF-1 signaling pathway and a functioning sulfur amino acid metabolism pathway, as the dramatic sickness of an *egl-9; suox-1* double mutant is suppressed by loss of *hif-1*, *cth-2*, or *cdo-1*. These genetic results validate the intersection of HIF-1 and CDO-1 in converging biological regulatory pathways and encourage further exploration of this regulatory node. The potential physiological relevance of the connection between HIF-1 signaling and cysteine metabolism is discussed below.

It was initially surprising that the canonical hypoxia-sensing transcription factor, that is activated by relatively low $O_2$ tensions, induces transcription of *cdo-1*, which encodes an oxidase that requires dissolved $O_2$ to oxidize cysteine. Both EGL-9 and CDO-1 are dioxygenases, requiring $O_2$ as a substrate to catalyze their chemical reactions. EGL-9 functions as an $O_2$ sensor, using its $O_2$ substrate to hydroxylate HIF-1 at relatively high $O_2$ partial pressures, targeting it for degradation. EGL-9 is uniquely poised to sense deviations from the normally high physiological $O_2$ concentrations given its high $K_m$ for $O_2$ (**Hirsilä et al., 2003**; **Dao et al., 2009**; **Ehrismann et al., 2007**; **Li et al., 2023**; **Losman et al., 2020**). While it is not known if mammalian CDO1 or *C. elegans* CDO-1 dioxygenases are active at lower $O_2$ tensions than EGL-9, it is possible, even probable, that the set point for activation of HIF-1 by the failure of EGL-9 prolyl hydroxylation, is at a higher $O_2$ tension than the $K_m$ of CDO-1 for oxidation of cysteine. In this way, CDO-1 could still coordinate cysteine catabolism while the cell is experiencing hypoxia and EGL-9 is unable to hydroxylate HIF-1.

## Evidence for distinct pathways of HIF-1 activation by hypoxia and cysteine/$H_2S$

The hypoxia-signaling pathway is defined by EGL-9-dependent prolyl hydroxylation of HIF-1. Hydroxylated HIF-1 is then targeted for degradation via VHL-1-mediated ubiquitination (**Epstein et al., 2001**; **Ivan et al., 2001**; **Kaelin and Ratcliffe, 2008**). However, multiple lines of evidence, reinforced by our work, demonstrate VHL-1- and prolyl hydroxylase-independent activity of EGL-9. *egl-9(-)* null mutant *C. elegans* accumulate HIF-1 protein and display increased transcription of many genes, including *nhr-57,* an established target of HIF-1 (**Shen et al., 2006**). In rescue experiments of an *egl-9(-)* null mutant, **Shao et al., 2009** demonstrate that a wild-type *egl-9* transgene restores normal HIF-1 protein levels and HIF-1 transcription. However, rescue experiments with a prolyl hydroxylase domain-inactive *egl-9(H487A)* transgene do not correct the accumulation of HIF-1 protein and only partially reduce the HIF-1 transcriptional output (**Shao et al., 2009**). Through our studies of *cdo-1* transcription, we demonstrate that an *egl-9(H487A)* mutant incompletely activates HIF-1 transcription when compared to an *egl-9(-)* null mutation (**Figure 5C and D**). Taken together, we conclude EGL-9 has activity independent of its prolyl hydroxylase domain, mirroring and supporting previous work (**Shao et al., 2009**).

In studies of the HIF-1-dependent P*nhr-57::GFP* transcriptional reporter, (**Shen et al., 2006**) observed that *egl-9(-)* null mutants promote HIF-1 transcription more than a *vhl-1(-)* mutant (**Shen et al., 2006**). We observe this same distinction between *egl-9* and *vhl-1* mutations with our P*cdo-1::GFP* reporter (**Figure 5C and D**). This difference in transcriptional activity is not explained by HIF-1 protein levels as HIF-1 protein accumulated equally in *egl-9(-)* and *vhl-1(-)* null mutants (**Shen et al., 2006**). This observation suggests that EGL-9 represses both HIF-1 levels and activity. This study also notes that *vhl-1* represses HIF-1 transcription in the intestine while *egl-9* acts in a wider array of tissues including the intestine and the hypodermis (**Shen et al., 2006**). **Budde and Roth, 2010** also

demonstrate that loss of *vhl-1* promotes HIF-1 transcription in the *C. elegans* intestine while total loss of *egl-9* promotes HIF-1 transcription in multiple tissues including the intestine and the hypodermis (*Budde and Roth, 2010*). Our data expand upon these observations by demonstrating that loss of the EGL-9 prolyl hydroxylase domain promotes *cdo-1* transcription in the intestine alone, mirroring the loss of *vhl-1* (*Figure 5C*). *Budde and Roth, 2010* additionally demonstrate that physiologically relevant stimuli also elicit a tissue-specific transcriptional response: hypoxia promotes HIF-1 transcription in the intestine while $H_2S$ promotes HIF-1 transcription in the hypodermis. Importantly, $H_2S$ promotes hypodermal HIF-1 transcription in a *vhl-1(-)* mutant, demonstrating a VHL-1-independent pathway for $H_2S$ activation of HIF-1 (*Budde and Roth, 2010*). We demonstrate that high levels of cysteine promote *cdo-1* transcription in the hypodermis. Taken together with our data, these studies suggest two distinct pathways for activating HIF-1 transcription: (i) a hypoxia-sensing pathway that is dependent upon *vhl-1* and the EGL-9 prolyl hydroxylase domain and promotes HIF-1 activity in the intestine and (ii) an $H_2S$-sensing pathway that is independent of *vhl-1* and the EGL-9 prolyl hydroxylase domain and promotes HIF-1 activity in the hypodermis.

The focus of CDO-1 expression and regulation of sulfite production in the hypoderm may be due to the demand in sulfur metabolism as well as oxygen-dependent hydroxylation of collagens during the *C. elegans* molting cycle. Many collagen genes are expressed before each larval molt and many collagen prolines are hydroxylated, like particular HIF-1 prolines, and their cysteines form disulfides during collagen assembly. The large demand for the amino acid cysteine in the many collagen genes expressed at high levels during a molt may challenge cysteine homeostasis in the hypodermis (*Meeuse et al., 2020*).

The genetic details of the $H_2S$-sensing pathway were solidified through studies of *rhy-1* in *C. elegans*. *Ma et al., 2012* demonstrate that *hif-1* repression via *rhy-1* requires the activity of *cysl-1*, a gene encoding a cysteine synthase-like protein (*Ma et al., 2012*). They further demonstrate that high $H_2S$ promotes a physical interaction between CYSL-1 and EGL-9, resulting in the inactivation of EGL-9 and increased HIF-1 activity. Together, these studies suggest distinct genetic regulators of EGL-9/HIF-1 signaling: *rhy-1* and *cysl-1* govern the $H_2S$-sensing pathway while *vhl-1* mediates the hypoxia-sensing pathway. These pathways are distinct in their requirement for the EGL-9 prolyl hydroxylase domain.

## A negative feedback loop senses high cysteine/$H_2S$, promotes CDO-1 activity, and maintains cysteine homeostasis

Why would the RHY-1/CYSL-1/EGL-9/HIF-1 $H_2S$-sensing pathway control the levels and activity of cysteine dioxygenase? We speculate this intersection facilitates a homeostatic pathway allowing *C. elegans* to sense and respond to cysteine level. We propose that $H_2S$ acts as a gaseous signaling molecule to promote cysteine catabolism. $H_2S$ activates HIF-1 in the hypodermis by promoting the CYSL-1-mediated inactivation of EGL-9 (*Ma et al., 2012*). We show that high levels of cysteine similarly induce *cdo-1* transcription in the hypodermis. Our genetic data demonstrate that *cdo-1* is induced by the same genetic pathway that senses $H_2S$ in *C. elegans* and CDO-1 acts in the hypodermis, the major site of $H_2S$-induced transcription. Furthermore, $H_2S$ induces endogenous *cdo-1* transcription more than threefold while *cdo-1* mRNA levels do not change when *C. elegans* are exposed to hypoxia (*Miller et al., 2011*; *Shen et al., 2005*). Thus, it is likely that $H_2S$ promotes *cdo-1* transcription through RHY-1, CYSL-1, EGL-9, and HIF-1. $H_2S$ is a reasonable small molecule signal to alert cells to high levels of cysteine. Excess cysteine results in the production of $H_2S$ mediated by multiple enzymes including cystathionase (CTH), cystathionine β-synthase (CBS), and 3-mercaptopyruvate sulfurtransferase (MST; *Singh and Banerjee, 2011*; *Jurkowska et al., 2014*). For example, CTH activity within the carotid body produces $H_2S$ that modifies the mammalian response to hypoxia (*Peng et al., 2010*). We speculate that excess cysteine in *C. elegans* promotes the enzymatic production of $H_2S$ which activates HIF-1 via the RHY-1/CYSL-1/EGL-9 signaling pathway. In our homeostatic model, $H_2S$-activated HIF-1 would then induce *cdo-1* transcription, promoting CDO-1 activity and the catabolism of the high-cysteine trigger (*Figure 6*). Supporting this model, *cysl-1(-)* and *hif-1(-)* mutant *C. elegans* cannot survive in a high cysteine environment, demonstrating their central role in promoting cysteine homeostasis.

Importantly, the human ortholog of EGL-9 (EglN1) has previously been implicated as a cysteine sensor in the context of triple negative breast cancer (TNBC) (*Briggs et al., 2016*). This study determined that HIF1α accumulates in TNBC cells even during normoxia. *Briggs et al., 2016* demonstrate

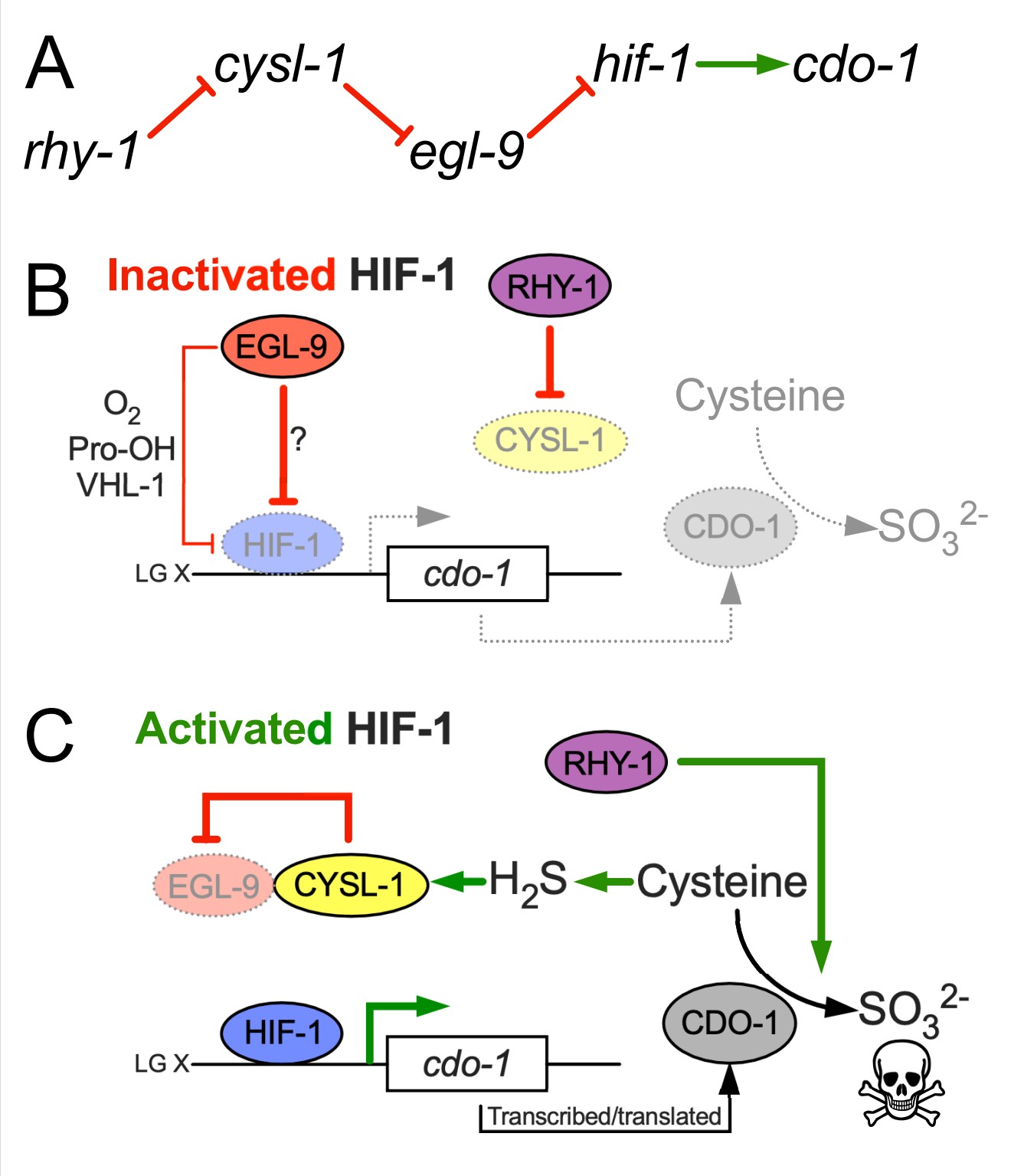

**Figure 6.** Model for the regulation of cysteine metabolism by HIF-1. (**A**) Proposed genetic pathway for the regulation of *cdo-1*. *rhy-1*, *cysl-1*, and *egl-9* act in a negative-regulatory cascade to control activity of the HIF-1 transcription factor, which activates transcription of *cdo-1*. (**B**) Under basal conditions (inactivated HIF-1), EGL-9 negatively regulates HIF-1 through 2 distinct pathways; one pathway is dependent upon O₂, prolyl hydroxylation (Pro-OH), and VHL-1, while the second acts independently of these canonical factors. Under these conditions *cdo-1* transcription is kept at basal levels and cysteine catabolism is not induced. (**C**) During conditions where HIF-1 is activated (high H₂S), CYSL-1 directly binds and inhibits EGL-9, preventing HIF-1 inactivation. Active HIF-1 binds the *cdo-1* promoter, driving transcription and promoting CDO-1 protein accumulation. High CDO-1 levels promote

*Figure 6 continued on next page*

*Figure 6 continued*

the catabolism of cysteine leading to production of sulfites ($SO_3^{2-}$) that are toxic during Moco or SUOX-1 deficiency. HIF-1-induced cysteine catabolism requires the activity of *rhy-1*.

that L-glutamate secretion via TNBC cells suppresses HIF1α prolyl hydroxylation, stabilizing HIF1α. L-glutamate secretion is mediated via the glutamate/cystine antiporter, xCT (**Bannai and Kitamura, 1980**; **Sato et al., 1999**). L-glutamate secretion inhibits xCT and a concomitant decrease in intracellular cystine/cysteine was observed. The authors propose that low intracellular cystine/cysteine produces oxidizing conditions that oxidize specific EglN1 cysteine residues, inhibiting the activity of EglN1. Thus, **Briggs et al., 2016** propose EglN1 as a cysteine sensor whose activity is promoted by cysteine.

Our work in *C. elegans* also strongly suggests a role for EGL-9 in sensing and responding to cysteine. We show that high levels of cysteine promote transcription of the HIF-1 target gene *cdo-1*, and that this induction is not additive with a null mutation in *egl-9,* suggesting cysteine and *egl-9* act in a pathway. Therefore, we propose that in *C. elegans* high levels of cysteine inhibit the activity of EGL-9, the opposite effect observed in TNBC cells. Furthermore, our genetic studies demonstrate that regulation of *cdo-1* transcription occurs largely independent of the EGL-9 prolyl hydroxylase domain and VHL-1, while the mechanism proposed by **Briggs et al., 2016** suggests that the stabilization of HIF1α by L-glutamate is correlated with increased prolyl hydroxylation. Despite these differences, it seems likely that the roles for *C. elegans* EGL-9 and human EglN1 in sensing cysteine are connected. However, additional studies are required to determine if there is an evolutionary relationship between these cysteine-sensing mechanisms.

Members of the $H_2S$-sensing pathway have also been implicated in the *C. elegans* response to infection (**Darby et al., 1999**; **Burton et al., 2021**). Many secreted proteins that mediate intercellular signaling or innate immune responses to pathogens are cysteine-rich and form disulfide bonds in the endoplasmic reticulum before protein secretion (**Gibbs et al., 2008**; **Frand et al., 2000**). Levels of free cysteine may fall during the massive inductions of cysteine-rich secreted proteins in development or immune defense, as well as during the synthesis of collagens in the periodic molting cycle (**Meeuse et al., 2020**; **Mizuki and Kasahara, 1992**). The oxidation of so many cysteines to disulfides in the endoplasmic reticulum might locally lower $O_2$ levels preventing EGL-9 hydroxylation of HIF-1. Thus, the intersection between HIF-1 and cysteine homeostasis we have uncovered may contribute to a regulatory axis in cell-cell signaling during development and in immune function.

## Methods
### General methods and strains

*C. elegans* were cultured using established protocols (**Brenner, 1974**). Briefly, animals were cultured at 20 °C on nematode growth media (NGM) seeded with wild-type *E. coli* (OP50). The wild-type strain of *C. elegans* was Bristol N2. Additional *E. coli* strains used in this work were BW25113 (Wild type, Moco+) and JW0764-2 (Δ*moaA753::kan,* Moco-; **Baba et al., 2006**).

*C. elegans* mutant and transgenic strains used in this work are listed here. When previously published, sources of strains are referenced. Unless a reference is provided, all strains were generated in this study.

### Non-transgenic *C. elegans*

ZG31, *hif-1(ia4) V* (**Jiang et al., 2001**)
JT307, *egl-9(sa307) V* (**Darby et al., 1999**)
GR2254, *moc-1(ok366) X* (**Warnhoff and Ruvkun, 2019**)
GR2260, *cdo-1(mg622) X* (**Warnhoff and Ruvkun, 2019**)
GR2261, *cdo-1(mg622) moc-1(ok366) X* (**Warnhoff and Ruvkun, 2019**)
GR2269, *suox-1(gk738847) X* (**Warnhoff and Ruvkun, 2019**)
CB5602, *vhl-1(ok161) X* (**Epstein et al., 2001**)
USD410, *cysl-1(ok762) X,* outcrossed 3 x for this work
USD414, *rhy-1(ok1402) II; suox-1(gk738847) X*

USD421, *egl-9(sa307) V; suox-1(gk738847) X*
USD422, *vhl-1(ok161) suox-1(gk738847) X*
USD430, *egl-9(sa307) V; cdo-1(mg622) suox-1(gk738847) X*
USD431, *hif-1(ia4) egl-9(sa307) V; suox-1(gk738847) X*
USD432, *rhy-1(ok1402) II; egl-9(sa307) V; suox-1(gk738847) X*
USD433, *cth-2(mg599) II; egl-9(sa307) V; suox-1(gk738847) X*
USD434, *egl-9(sa307) X; cysl-1(ok762) suox-1(gk738847) X*
USD512, *rhy-1(ok1402) II,* outcrossed 4 x for this work
USD706, *unc-119(ed3) III; cdo-1(mg622) moc-1(ok366) X*
USD920, *cdo-1(rae273) moc-1(ok366) X*
USD921, *egl-9(sa307) V; cdo-1(rae273) X*
USD922, *rhy-1(ok1402) II; cdo-1(rae273) X*
USD937, *egl-9(rae276) V; suox-1(gk738847) X*

## MiniMos transgenic lines

USD531, *unc-119(ed3) III; raeTi1 [Pcdo-1::CDO-1::GFP unc-119(+)]*
USD719, *unc-119(ed3) III; cdo-1(mg622) moc-1(ok366); raeTi14 [Pcdo-1::CDO-1(C85Y)::GFP unc-119(+)]*
USD720, *unc-119(ed3) III; raeTi15 [Pcdo-1::GFP unc-119(+)]*
USD730, *rhy-1(ok1402) II; unc-119(ed3) III; raeTi15 [Pcdo-1::GFP unc-119(+)]*
USD733, *unc-119(ed3) III; egl-9(sa307) V; raeTi15 [Pcdo-1::GFP unc-119(+)]*
USD739, *unc-119(ed3) III; cdo-1(mg622) moc-1(ok366) X; raeTi1 [Pcdo-1::CDO-1::GFP unc-119(+)]*
USD766, *unc-119(ed3) III; cdo-1(mg622) moc-1(ok366) X; raeTi32 [Pcol-10::CDO-1::GFP unc-119(+)]*
USD767, *unc-119(ed3) III; cdo-1(mg622) moc-1(ok366) X; raeTi33 [Pcol-10::CDO-1::GFP unc-119(+)]*
USD776, *rhy-1(ok1402) II; unc-119(ed3) III; hif-1(ia4) V; raeTi15 [Pcdo-1::GFP unc-119(+)]*
USD777, *unc-119(ed3) III; egl-9(sa307) hif-1(ia4) V; raeTi15 [Pcdo-1::GFP unc-119(+)]*
USD780, *rhy-1(ok1402) II; unc-119(ed3) III; cysl-1(ok762) X; raeTi15 [Pcdo-1::GFP unc-119(+)]*
USD787, *unc-119(ed3) III; egl-9(sa307) V; cysl-1(ok762) X; raeTi15 [Pcdo-1::GFP unc-119(+)]*
USD808, *unc-119(ed3) III; cdo-1(mg622) moc-1(ok366) X; raeTi40 [Pcol-10::CDO-1[C85Y]::GFP unc-119(+)]*
USD810, *unc-119(ed3) III; cdo-1(mg622) moc-1(ok366) X; raeTi41 [Pcol-10::CDO-1[C85Y]::GFP unc-119(+)]*
USD940, *unc-119(ed3) III; vhl-1(ok161) X; raeTi15 [Pcdo-1::GFP unc-119(+)]*
USD1160, *unc-119(ed3) III; cysl-1(ok762) X; raeTi15 [Pcdo-1::GFP unc-119(+)]*
USD1161, *unc-119(ed3) III; hif-1(ia4) V; raeTi15 [Pcdo-1::GFP unc-119(+)]*

## EMS-derived strains

USD659, *unc-119(ed3) III; egl-9(rae213) V; raeTi1 [Pcdo-1::CDO-1::GFP unc-119(+)]*
USD674, *unc-119(ed3) III; egl-9(rae227) V; raeTi1 [Pcdo-1::CDO-1::GFP unc-119(+)]*
USD655, *rhy-1(rae209) II; unc-119(ed3) III; raeTi1 [Pcdo-1::CDO-1::GFP unc-119(+)]*
USD656, *rhy-1(rae210) II; unc-119(ed3) III; raeTi1 [Pcdo-1::CDO-1::GFP unc-119(+)]*
USD657, *rhy-1(rae211) II; unc-119(ed3) III; raeTi1 [Pcdo-1::CDO-1::GFP unc-119(+)]*
USD658, *rhy-1(rae212) II; unc-119(ed3) III; raeTi1 [Pcdo-1::CDO-1::GFP unc-119(+)]*

## CRISPR/Cas9-derived strains

USD914, *cdo-1(rae273) X,* CDO-1::GFP
USD926, *egl-9(rae276) V,* EGL-9[H487A]
USD928, *unc-119(ed3) III; egl-9(rae278) V; raeTi15 [Pcdo-1::GFP unc-119(+)]*

## MiniMos transgenesis

Cloning of original plasmid constructs was performed using isothermal/Gibson assembly (*Gibson et al., 2009*). All MiniMos constructs were assembled in pNL43, which is derived from pCFJ909, a gift from Erik Jorgensen (Addgene plasmid #44480; *Lehrbach and Ruvkun, 2016*). Details about plasmid construction are described below. MiniMos transgenic animals were generated using established protocols that rescue the *unc-119(ed3)* Unc phenotype (*Frøkjær-Jensen et al., 2014*).

To generate a construct that expressed CDO-1 under the control of its native promoter, we cloned the wild-type *cdo-1* genomic locus from 1,335 base pairs upstream of the *cdo-1* ATG start codon to (and including) codon 190 encoding the final CDO-1 amino acid prior to the TAA stop codon. This wild-type genomic sequence was fused in frame with a C-terminal GFP and *tbb-2* 3'UTR (376 bp downstream of the *tbb-2* stop codon) in the pNL43 plasmid backbone. This plasmid is called pKW24 (P*cdo-1::CDO-1::GFP*).

To generate pKW44, a construct encoding the active site mutant transgene *Pcdo-1::CDO-1(C85Y)::GFP*, we performed Q5 site-directed mutagenesis on pKW24, following manufacturer's instructions (New England Biolabs). In pKW44, codon 85 was mutated from TGC (cysteine) to TAC (tyrosine).

To generate pKW45 (P*cdo-1::GFP*), the 1,335 base pair *cdo-1* promoter was amplified and fused directly to the GFP coding sequence. Both fragments were derived from pKW24, excluding the *cdo-1* coding sequence and introns.

pKW49 is a construct driving *cdo-1* expression from the hypodermal-specific *col-10* promoter (*Pcol-10::CDO-1::GFP*) (*Hong et al., 2000*). The *col-10* promoter (1,126 base pairs upstream of the *col-10* start codon) was amplified and fused upstream of the *cdo-1* ATG start codon in pKW24, replacing the native *cdo-1* promoter. pKW53 [P*col-10::CDO-1(C85Y)::GFP*] was engineered using the same Q5-site-directed mutagenesis strategy as was described for pKW44. However, this mutagenesis used pKW49 as the template plasmid.

## Chemical mutagenesis and whole genome sequencing

To define *C. elegans* gene activities that were necessary for the control of *cdo-1* levels, we carried out a chemical mutagenesis screen for animals that accumulate CDO-1 protein. To visualize CDO-1 levels, we engineered USD531, a transgenic *C. elegans* strain carrying the *raeTi1* [pKW24, *Pcdo-1::CDO-1::GFP*] transgene. USD531 transgenic *C. elegans* were mutagenized with ethyl methanesulfonate (EMS) using established protocols (*Brenner, 1974*). F2 generation animals were manually screened, and mutant isolates were collected that displayed high accumulation of *Pcdo-1::CDO-1::GFP*. We demanded that mutant strains of interest were viable and fertile.

We followed established protocols to identify EMS-induced mutations in our strains of interest (*Lehrbach et al., 2017*). Briefly, whole genomic DNA was prepared from *C. elegans* using the Gentra Puregene Tissue Kit (Qiagen) and genomic DNA libraries were prepared using the NEBNext genomic DNA library construction kit (New England Biolabs). DNA libraries were sequenced on an Illumina Hi-Seq and deep sequencing reads were analyzed using standard methods on Galaxy, a web-based platform for computational analyses (*Galaxy Community, 2022*). Briefly, sequencing reads were trimmed and aligned to the WBcel235 *C. elegans* reference genome (*Bolger et al., 2014*; *Li and Durbin, 2010*). Variations from the reference genome and the putative impact of those variations were annotated and extracted for analysis (*Wilm et al., 2012*; *Cingolani et al., 2012a*; *Cingolani et al., 2012b*). Here, we report the analysis of 6 new mutant strains (USD655, USD656, USD657, USD658, USD659, and USD674). Among these mutant strains, we found two unique mutations in *egl-9* (USD659 and USD674) and four unique mutations in *rhy-1* (USD655, USD656, USD657, USD658). The allele names and molecular identity of these new *egl-9* and *rhy-1* mutations are specified in *Figure 1C*. These genes were prioritized based on the isolation of multiple independent alleles and their established functions in a common pathway, the hypoxia and H$_2$S-sensing pathway (*Shen et al., 2006*; *Budde and Roth, 2011*; *Ma et al., 2012*). Whole genome sequencing data for these *C. elegans* strains have been deposited at the NIH Sequence Read Archive (SRA) under accession PRJNA1063314.

## Genome engineering by CRISPR/Cas9

We followed standard protocols to perform CRISPR/Cas9 genome engineering of *cdo-1* and *egl-9* genomic loci in *C. elegans* (*Ghanta and Mello, 2020*; *Arribere et al., 2014*; *Paix et al., 2017*; *Dokshin*

*et al., 2018*). Essential details of the CRISPR/Cas9-generated reagents in this work are described below.

We used homology-directed repair to generate *cdo-1(rae273)* [CDO-1::GFP]. The guide RNA was 5'-gactacagaggatctaagaa-3' (crRNA, IDT). The GFP donor double-stranded DNA (dsDNA) was amplified from pCFJ2249 using primers that contained roughly 40 bp of homology to *cdo-1* flanking the desired insertion site (*Aljohani et al., 2020*). The primers used to generate the donor dsDNA were: 5'-gtacggcaagaaagttgactacagaggatctaagaataatagtactagcggtggcagtgg-3' and 5'-agaatcaacacgttat tacattgagggatatgttgtttacttgtagagctcgtccattcc-3'. Glycine 189 of CDO-1 was removed by design to eliminate the PAM site and prevent cleavage of the donor dsDNA.

The *egl-9(rae276)* and *egl-9(rae278)* [EGL-9(H487A)] alleles were also generated by homology-directed repair using the same combination of guide RNA and single-stranded oligodeoxynucleotide (ssODN) donor. The guide RNA was 5'-tgtgaagcatgtagataatc-3' (crRNA, IDT) The ssODN donor was 5'-gcttgccatctatcctggaaatggaactcgttatgtgaaggctgtagacaatccagtaaaagatggaagatgtataaccactat ttattactg-3' (Ultramer, IDT). Successful editing resulted in altering the coding sequence of EGL-9 to encode for an alanine rather than the catalytically essential histidine at position 487. We also used synonymous mutations to introduce an AccI restriction site that is helpful for genotyping the *rae276* and *rae278* mutant alleles.

## *C. elegans* growth assays

To assay developmental rates, *C. elegans* were synchronized at the first stage of larval development. To synchronize animals, embryos were harvested from gravid adult animals via treatment with a bleach and sodium hydroxide solution. Embryos were then incubated overnight in M9 solution causing them to hatch and arrest development at the L1 stage (*Stiernagle, 2006*). Synchronized L1 animals were cultured on NGM seeded with BW25113 (Wild type, Moco+) or JW0764-2 (Δ*moaA753::kan*, Moco-) *E. coli*. Animals were cultured for 48 or 72 hr (specified in the appropriate figure legends), and live animals were imaged as described below. Animal length was measured from tip of head to the end of the tail.

To determine qualitative 'health' of various *C. elegans* strains, we assayed the ability of these strains to consume all *E. coli* food provided on an NGM petri dish. For this experiment, dietary *E. coli* was produced via overnight culture in liquid LB in a 37 °C shaking incubator. A total of 200 μl of this *E. coli* was seeded onto NGM petri dishes and allowed to dry, producing nearly identical lawns and growth environments. Then, 5 L4 animals of a strain of interest were introduced onto these NGM petri dishes seeded with OP50. For each experiment, petri dishes were monitored daily and scored when all *E. coli* was consumed by the population of animals. This assay is beneficial because it integrates many life-history measures (i.e. developmental rate, brood size, embryonic viability, etc.) into a single simple assay that can be scaled and applied to many *C. elegans* strains in parallel.

## Cysteine exposure

To determine the impact of supplemental cysteine on expression of *cdo-1* and animal viability, we exposed various *C. elegans* strains to 0, 50, 100, 250, 500, or 1000 μM supplemental cysteine. *C. elegans* strains were synchronously grown on NGM media supplemented with *E. coli* OP50 at 20 °C until reaching the L4 stage of development. Live L4 animals were then collected from the petri dishes and cultured in liquid M9 media containing 4 X concentrated *E. coli* OP50 with or without supplemental cysteine. These liquid cultures were gently rocked at 20 °C overnight. Post exposure, GFP imaging was performed as described in the Microscopy section of the materials and methods. Alternatively, animals exposed overnight to 0, 100, or 1000 μM supplemental cysteine were scored for viability after being seeded onto NGM petri dishes. Animals were determined to be alive if they responded to mechanical stimulus.

## Microscopy

Low-magnification bright field and fluorescence images (imaging GFP simultaneously in multiple animals) were collected using a Zeiss AxioZoom V16 microscope equipped with a Hamamatsu Orca flash 4.0 digital camera using Zen software (Zeiss). For experiments with supplemental cysteine, low magnification bright field and fluorescence images were collected using a Nikon SMZ25 microscope equipped with a Hamamatsu Orca flash 4.0 digital camera using NIS-Elements software (Nikon). High

magnification differential interference contrast (DIC) and GFP fluorescence images (imaging CDO-1::GFP encoded by *cdo-1(rae273)*) were collected using Zeiss AxioImager Z1 microscope equipped with a Zeiss AxioCam HRc digital camera using Zen software (Zeiss). All images were processed and analyzed using ImageJ software (NIH). All imaging was performed on live animals paralyzed using 20 mM sodium azide. For all fluorescence images shown within the same figure panel, images were collected using the same exposure time and processed identically. To quantify GFP expression, the average pixel intensity was determined within a set transverse section immediately anterior to the developing vulva. Background pixel intensity was determined in a set region of interest distinct from the *C. elegans* samples and was subtracted from the sample measurements.

## Acknowledgements

Some *C. elegans* strains were provided by the CGC, which is funded by the NIH Office of Research Infrastructure Programs (P40 OD010440). Research reported in this publication was supported by the National Institute of General Medical Sciences of the National Institutes of Health under award numbers R01 GM044619 (to GR) and R35 GM146871 (to KW).

## Additional information

### Funding

| Funder | Grant reference number | Author |
|---|---|---|
| National Institute of General Medical Sciences | R35 GM146871 | Kurt Warnhoff |
| National Institute of General Medical Sciences | R01 GM044619 | Gary Ruvkun |

The funders had no role in study design, data collection and interpretation, or the decision to submit the work for publication.

### Author contributions

Kurt Warnhoff, Conceptualization, Data curation, Supervision, Funding acquisition, Investigation, Methodology, Writing – original draft, Writing – review and editing; Sushila Bhattacharya, Jennifer Snoozy, Peter C Breen, Investigation, Methodology; Gary Ruvkun, Conceptualization, Supervision, Funding acquisition, Writing – original draft, Writing – review and editing

### Author ORCIDs

Kurt Warnhoff ⬡ https://orcid.org/0000-0002-9503-0557
Gary Ruvkun ⬡ https://orcid.org/0000-0002-7473-8484

Reviewer #2 (Public Review): https://doi.org/10.7554/eLife.89173.3.sa1
Reviewer #3 (Public Review): https://doi.org/10.7554/eLife.89173.3.sa2
Reviewer #4 (Public Review): https://doi.org/10.7554/eLife.89173.3.sa3
Author Response https://doi.org/10.7554/eLife.89173.3.sa4

## Additional files

### Supplementary files
• MDAR checklist

### Data availability

All C. elegans strains, bacterial strains, and plasmids are described in the Methods section and are available from the corresponding authors with no restrictions. Source data have been deposited on Dryad and can be accessed at https://doi.org/10.5061/dryad.kd51c5bdk. Whole genome sequencing data have been deposited at the NIH BioProject under accession PRJNA1063314.

The following datasets were generated:

| Author(s) | Year | Dataset title | Dataset URL | Database and Identifier |
|---|---|---|---|---|
| Warnhoff K, Bhattacharya S, Snoozy J, Breen P, Ruvkun G | 2024 | Hypoxia-inducible factor induces cysteine dioxygenase and promotes cysteine homeostasis in *Caenorhabditis elegans* | https://www.ncbi.nlm.nih.gov/bioproject/PRJNA1063314 | NCBI BioProject, PRJNA1063314 |
| Warnhoff K, Bhattacharya S, Snoozy J, Breen P, Ruvkun G | 2024 | Data from: Hypoxia-inducible factor induces cysteine dioxygenase and promotes cysteine homeostasis in *Caenorhabditis elegans* | https://doi.org/10.5061/dryad.kd51c5bdk | Dryad Digital Repository, 10.5061/dryad.kd51c5bdk |

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
