## [Editor Report · eLife assessment]

The study presents **valuable** findings on how the hypoxia response pathway senses and responds to changes in the homeostasis of the amino acid cysteine and other sulfur-containing molecules. By providing a **compelling**, rigorous genetic analysis of the pathway, the study adds to a growing body of literature showing that prolyl hydroxylation is not the only mechanism by which the hypoxia response pathway can act. Although the paper does not reveal new biochemical insight into the mechanism, it opens up new areas of investigation that will be of interest to cell biologists and biomedical researchers studying the many pathologies involving hypoxia and/or cysteine metabolism.

---

## [Referee Report · Reviewer #2 (Public Review)]

The authors investigate the transcriptional regulation of cysteine dioxygenase (CDO-1) in *C. elegans* and its role in maintaining cysteine homeostasis. They show that high cysteine levels activate cdo-1 transcription through the hypoxia-inducible transcription factor HIF-1. Using transcriptional and translational reporters for CDO-1, the authors propose that a negative feedback pathway involving RHY-1, CYSL-1, EGL-9 and HIF-1 in regulating cysteine homeostasis.

Genetics is a notable strength of this study. The forward genetic screen, gene interaction and epistasis analyses are beautifully designed and rigorously conducted, yielding solid and unambiguous conclusions on the genetic pathway regulating CDO-1. The writing is clear and accessible, contributing to the overall high quality of the manuscript.

Addressing the specifics of cysteine supplementation and interpretation regarding the cysteine homeostasis pathway would further clarify the paper and strengthen the study's conclusions.

First, the authors show that the supplementation of exogenous cysteine activates cdo-1p::GFP. Rather than showing data for one dose, the author may consider presenting dose-dependency results and whether cysteine activation of cdo-1 also requires HIF-1 or CYSL-1, which would be important data given the focus and major novelty of the paper in cysteine homeostasis, not the cdo-1 regulatory gene pathway. While the genetic manipulation of cdo-1 regulators yields much more striking results, the effect size of exogenous cysteine is rather small. Does this reflect a lack of extensive condition optimization or robust buffering of exogenous/dietary cysteine? Would genetic manipulation to alter intracellular cysteine or its precursors yield similar or stronger effect sizes?

Second, there remain several major questions regarding the interpretation of the cysteine homeostasis pathway. How much specificity is involved for the RHY-1/CYSL-1/EGL-9/HIF-1 pathway to control cysteine homeostasis? Is the pathway able to sense cysteine directly or indirectly through its metabolites or redox status in general? Given the very low and high physiological concentrations of intracellular cysteine and glutathione (GSH, a major reserve for cysteine), respectively, there is a surprising lack of mention and testing of GSH metabolism. In addition, what are the major similarities and differences of cysteine homeostasis pathways between *C. elegans* and other systems (HIF dependency, transcription vs post-transcriptional control)? These questions could be better discussed and noted with novel findings of the current study that are likely *C. elegans* specific or broadly conserved.

All of my comments and questions above have been satisfactorily addressed in the revised manuscript.

---

## [Referee Report · Reviewer #3 (Public Review)]

There has been a long-standing link between the biology of sulfur-containing molecules (e.g., hydrogen sulfide gas, the amino acid cysteine, and its close relative cystine, et cetera) and the biology of hypoxia, yet we have a poor understanding of how and why these two biological processes and are co-regulated. Here, the authors use *C. elegans* to explore the relationship between sulfur metabolism and hypoxia, examining the regulation of cysteine dioxygenase (CDO1 in humans, CDO-1 in *C. elegans*), which is critical to cysteine catabolism, by the hypoxia inducible factor (HIF1 alpha in humans, HIF-1 in *C. elegans*), which is the key terminal effector of the hypoxia response pathway that maintains oxygen homeostasis. The authors are trying to demonstrate that (1) the hypoxia response pathway is a key regulator of cysteine homeostasis, specifically through the regulation of cysteine dioxygenase, and (2) that the pathway responds to changes in cysteine homeostasis in a mechanistically distinct way from how it responds to hypoxic stress.

Briefly summarized here, the authors initiated this study by generating transgenic animals expressing a CDO-1::GFP protein chimera from the cdo-1 promoter so that they could identify regulators of CDO-1 expression through a forward genetic screen. This screen identified mutants with elevated CDO-1::GFP expression in two genes, egl-9 and rhy-1, whose wild-type products are negative regulators of HIF-1, raising the possibility that cdo-1 is a HIF-1 transcriptional target. Indeed, the authors provide data showing that cdo-1 regulation by EGL-9 and RHY-1 is dependent on HIF-1 and that regulation by RHY-1 is dependent on CYSL-1, as expected from other published findings of this pathway. The authors show that exogenous cysteine activates cdo-1 expression, reflective of what is known to occur in other systems. Moreover, they find that exogenous cysteine is toxic to worms lacking CYSL-1 or HIF-1 activity, but not CDO-1 activity, suggesting that HIF-1 mediates a survival response to toxic levels of cysteine and that this response requires more than just the regulation of CDO-1. The authors validate their expression studies using a GFP knockin at the cdo-1 locus, and they demonstrate that a key site of action for CDO-1 is the hypodermis. They present genetic epistasis analysis supporting a role for RHY-1, both as a regulator of HIF-1 and as a transcriptional target of HIF-1, in offsetting toxicity from aberrant sulfur metabolism. The authors use CRISPR/Cas9 editing to mutate a key amino acid in the prolyl hydroxylase domain of EGL-9, arguing that EGL-9 inhibits CDO-1 expression through a mechanism that is largely independent of the prolyl hydroxylase activity.

Overall, the data seem rigorous, and the conclusions drawn from the data seem appropriate. The experiments test the hypothesis using logical and clever molecular genetic tools and design. The sample size is a bit lower than is typical for *C. elegans* papers; however, the experiments are clearly not underpowered, so this is not an issue. The paper is likely to drive many in the field (including the authors themselves) into deeper experiments on (1) how the pathway senses hypoxia and sulfur/cysteine/H2S using these distinct mechanisms/modalities, (2) how oxygen and sulfur/cysteine/H2S homeostasis influence one another, and (3) how this single pathway evolved to sense and respond to both of these stress modalities.

My previous concerns have been addressed. The authors are commended on an excellent body of research.

---

## [Referee Report · Reviewer #4 (Public Review)]

Summary:

This is a revised manuscript that describes a role for cdo-1 in regulating cellular cysteine levels. The authors show that expression of cdo-1, predicted to encode a cysteine dioxygenase, is regulated by HIF-1, the conserved hypoxia-induced transcription factor. The expression of cdo-1 is controlled by the RHY-1/CYSL-1/EGL-9/HIF-1 pathway that has been demonstrated to be involved in the response to H2S.

Strengths:

The new finding of this study is that cdo-1, predicted to encode a cysteine dioxygenase, is expressed in the hypodermis and that hypodermal expression rescues at least one phenotype of the cdo-1(mg622) mutant (ability to survive toxic sulfite accumulation in Moco-deficient conditions). Using sulfite toxicity is an interesting reporter for cellular cysteine abundance.

Weaknesses:

The authors claim more than once that the H2S/Cys responsive pathway is RHY-1 - CYSL-1 - EGL-9 - HIF-1. Their data don't seem to support this claim, as they show that Pcdo-1::GFP is induced in rhy-1 mutants incubated with cysteine. It is therefore not appropriate to claim that "HIF-1-induced cysteine catabolism requires the activity of rhy-1" that they include in the description of the model in Fig 6. There is simply no evidence at all that RHY-1 has any role in modulating the activity of CDO-1 other than through transcriptional activation via HIF-1.

I don't find the arguments that this pathway is required for cysteine homeostasis per se (as claimed in the last sentence of the introduction). The authors expose worms to excess cysteine for 48 hours in liquid culture with bacteria. It is well known in these conditions that the bacteria will produce H2S from the cysteine in the culture. All of the cysteine exposure data shown can be explained by the effect of H2S exposure. This would explain why hif-1 and cysl-1 mutants die but cdo-1 mutants do not, for example. The authors don't provide any data to rule out the possibility that bacterial H2S production underlies these results. This explains why the pathway described in this work is the same as has been previously described. Similarly, there is no evidence at all to support their assertion that there are "other pathways" induced by HIF-1 to deal with sulfite produced by cysteine catabolism. However, if the main problem is H2S production (perhaps by bacteria) then cdo-1 would not be relevant and the mutants would be viable as observed.

In a couple of places, the authors seem to argue that H2S-induced expression is limited to the hypodermis and hypoxia-induced gene expression is mostly in the intestine. This is consistent with the expression of cdo-1 (this work) and nhr-57 (Budde and Roth) but it is not appropriate to generalize this. Previous work from the Ruvkun lab (Ma et al) show that the CYSL-1 regulates expression of HIF-1 targets in neurons. Moreover, HIF-1 protein accumulates in the nucleus of nearly all cells, and there is no reason to believe that there are changes in the expression of other genes in different tissues.

---

## [Author Response]

The following is the authors’ response to the original reviews.

**Reviewer #1 (Public Review):**
Issue 1: The relevance is somewhat unclear. High cysteine levels can be achieved in the laboratory, but, is this relevant in the life of *C. elegans*? Or is there physiological relevance in humans, e.g. a disease? The authors state "cells and animals fed excess cysteine and methionine", but is this more than a laboratory excess condition? SUOX nonfunctional conditions in humans don't appear to tie into this, since, in that context, the goal is to inactivate CDO or CTH to prevent sulfite production. The authors also mention cancer, but the link to cysteine levels is unclear. In that sense, then, the conditions studied here may not carry much physiological relevance.

Response 1: We set out to answer a fundamental question: what pathways regulate the function of cysteine dioxygenase, a highly conserved enzyme in sulfur amino acid metabolism? In an unbiased genetic screen that sampled millions of EMS generated mutations across all ~20,000 *C. elegan*s genes, we discovered loss of function/null mutations in egl-9 and rhy-1, two negative regulators of the hypoxia inducible transcription factor (hif-1). Genetic ablation of the egl-9 or rhy-1 loci are likely not relevant to the life of a *C. elegans* animal, i.e. this is not representative of a natural state. Yet, this extreme genetic intervention has taught us a new fundamental truth about the interaction between EGL-9/RHY-1, HIF-1, and the transcriptional activation of cdo1. Similarly, the high cysteine levels used in our assays may or may not be representative of a state in nature, we do not know (nor do we make any claims about the environmental relevance of our choice of cysteine concentrations). It seems very plausible that pathological states exist where cysteine concentrations may rise to comparable levels in our experimental system. More importantly, we have started with excess to physiology to elicit a clear response that we can study in the lab. Similar strategies established the cysteine-induction phenotype of CDO1 in mammalian systems. For instance, in Kwon and Stipanuk 2001, hepatocytes are cultured in media supplemented with 2mmol/L cysteine to promote a ~4-fold increase in CDO1 mRNA.

Issue 2: The pathway is described as important for cysteine detoxification, which is described to act via H2S (Figure 6). Much of that pathway has already been previously established by the Roth, Miller, and Horvitz labs as critical for the H2S response. While the present manuscript adds some additional insight such as the additional role of RHY-1 downstream on HIF-1 in promoting toxicity, this study therefore mainly confirms the importance of a previously described signalling pathway, essentially adding a new downstream target rhy-1 -> cysl-1 -> egl-9 -> hif-1 -> sqrd-1/cdo-1. The impact of this finding is reduced by the fact that cdo-1 itself isn't actually required for survival in high cysteine, suggesting it is merely a maker of the activity of this previously described pathway.

Response 2: We agree that the primary impact of our manuscript is the establishment of a novel intersection between the H2S-sensing pathway (largely worked out by Roth, Miller, and Horvitz) and our gene of interest, cysteine dioxygenase. We believe that the connection between these two pathways is exciting as it suggests a logical homeostatic circuit. High cysteine yields enzymatically produced H2S. This H2S may then act as a signal promoting HIF-1 activity (via RHY-1/CYSL-1/EGL-9). High HIF-1 activity increases cdo-1 transcription and activity promoting the degradation of the high-cysteine trigger. As pointed out by the reviewer, cdo-1(-) loss of function alone does not cause cysteine sensitivity at the concentrations tested. Given that cysl-1(-) and hif-1(-) mutants are exquisitely sensitive to high levels of cysteine, we propose that HIF-1 activates the transcription of additional genes that are required for high cysteine tolerance. However, our genetic data show that cdo-1 is more than simply a marker of HIF-1 transcription. Our genetic data in Table 1 demonstrate that HIF-1 activation (caused by egl-9(-)) is sufficient to cause severe sickness in a suox-1 hypomorphic mutant which cannot detoxify sulfites, a critical product of cysteine catabolism. This severe sickness can be reversed by inactivating hif-1, cth-2, or cdo-1. These data demonstrate a functional intersection between the established H2S-sensing pathway and cysteine catabolism governed by cdo-1.

**Reviewer #2 (Public Review):**
Issue 3: First, the authors show that the supplementation of exogenous cysteine activates cdo-1p::GFP. Rather than showing data for one dose, the author may consider presenting dose-dependency results and whether cysteine activation of cdo-1 also requires HIF-1 or CYSL-1, which would be important data given the focus and major novelty of the paper in cysteine homeostasis, not the cdo-1 regulatory gene pathway.

Response 3: We agree with the reviewer and have performed the suggested dose-response curve for expression of Pcdo-1::GFP in wild-type *C. elegans*. We observe substantial activation of the Pcdo-1::GFP transcriptional reporter beginning at 100µM supplemental cysteine (Figure 3C). Higher doses of cysteine do not elicit a substantially stronger induction of the Pcdo-1::GFP reporter. Thus, we find that 100µM supplemental cysteine strikes the right balance between strongly inducing the Pcdo-1::GFP reporter while not inducing any toxicity or lethality in wild-type animals (Figure 3E).

We further agree that testing for induction of the Pcdo-1::GFP reporter in a hif-1(-) or cysl-1(-) mutant background is a critical experiment. However, we have not been able to identify a cysteine concentration that induces Pcdo-1::GFP and is not 100% lethal for hif-1(-) or cysl-1(-) mutant *C. elegans*. The remarkable sensitivity of hif-1(-) or cysl-1(-) mutant *C. elegans* to supplemental cysteine demonstrates the critical role of these genes in promoting cysteine homeostasis. But because of this lethality, we could not assay the Pcdo1::GFP reporter in the hif-1(-) or cysl-1(-) mutant animals. But the lethality to excess cysteine demonstrates that this cysteine response is salient. To get at how cysteine might be interacting with the HIF-1-signaling pathway, we performed new additivity experiments by supplementing 100µM cysteine to wild type, egl-9(-), and rhy-1(-) mutant *C. elegans* expressing the Pcdo-1::GFP reporter. Surprisingly, we found that cysteine had no significant impact on Pcdo-1::GFP expression in an egl-9(-) mutant background but significantly increased the Pcdo-1::GFP expression in a rhy-1(-) background (Figure 3A,B). These data suggest that cysteine acts in a pathway with egl-9 and in parallel to rhy-1. These data have been incorporated into Figure 3A,B and are included in the Results section of the manuscript.

Issue 4: While the genetic manipulation of cdo-1 regulators yields much more striking results, the effect size of exogenous cysteine is rather small. Does this reflect a lack of extensive condition optimization or robust buffering of exogenous/dietary cysteine? Would genetic manipulation to alter intracellular cysteine or its precursors yield similar or stronger effect sizes?

Response 4: We agree that the induction of the Pcdo-1::GFP reporter by supplemental cysteine is not as dramatic as the induction caused by the egl-9 or rhy-1 null alleles. We believe our Response 3 and new Figure 3C demonstrate that this phenomenon is not due to lack of condition optimization, but likely reflects some biology. As pointed out by the reviewer, *C. elegans* likely buffers exogenous cysteine and this (perhaps) prevents the impressive Pcdo-1::GFP induction observed in the egl-9(-) and rhy-1(-) mutant animals. We have now mentioned this possible interpretation in the Results section. Furthermore, we like the idea of using genetic tricks to promote cysteine accumulation within *C. elegans* cells and tissues and will consider these approaches in future studies.

Issue 5: Second, there remain several major questions regarding the interpretation of the cysteine homeostasis pathway. How much specificity is involved for the RHY-1/CYSL-1/EGL-9/HIF-1 pathway to control cysteine homeostasis? Is the pathway able to sense cysteine directly or indirectly through its metabolites or redox status in general? Given the very low and high physiological concentrations of intracellular cysteine and glutathione (GSH, a major reserve for cysteine), respectively, there is a surprising lack of mention and testing of GSH metabolism.

Response 5: Future studies are required to determine the specificity of the RHY-1/CYSL-1/EGL-9/HIF-1 pathway for the control of cysteine homeostasis. Our proposed mechanism, that H2S activates the HIF-1 pathway is based largely on the work of the Horvitz lab (Ma et al. 2012). They demonstrate that H2S promotes a direct inhibitory interaction between CYSL-1 and EGL-9, leading to activation of HIF-1. These findings align nicely with our genetic and pharmacological data. However, our work does not provide direct evidence as to the cysteine-derived metabolite that activates HIF-1. We propose H2S as a likely candidate.

We have added a note to the introduction regarding the role of GSH as a reservoir of excess cysteine and agree that future studies might find interesting links between CDO-1, GSH metabolism, and HIF-1.

Issue 6: In addition, what are the major similarities and differences of cysteine homeostasis pathways between *C. elegans* and other systems (HIF dependency, transcription vs post-transcriptional control)? These questions could be better discussed and noted with novel findings of the current study that are likely *C. elegans* specific or broadly conserved.

Response 6: We have included a new section in the Discussion highlighting the nature of mammalian CDO1 regulation. We propose the hypothesis that a homologous pathway to the *C. elegans* RHY-1/CYSL-1/EGL9/HIF-1 pathway might operate in mammalian cells to sense high cysteine and induce CDO1 transcription. Importantly, all proteins in the *C. elegans* pathway have homologous counterparts in mammals. However, this hypothesis remains to be tested in mammalian systems.

**Reviewer #3 (Public Review):**
Major weaknesses of the paper include:Issue 7: the over-reliance on genetic approaches.

Response 7: This is a fair critique. Our expertise is genetics. Our philosophy, which the reviewers may not share, is that there is no such thing as too much genetics!

Issue 8: the lack of novelty regarding prolyl hydroxylase-independent activities of EGL-9.

Response 8: We believe the primary novelty of our work is establishing the intersection between the H2Ssensing HIF-1 pathway and cysteine catabolism governed by cysteine dioxygenase. Our demonstration that cdo-1 regulation operates largely independent of VHL-1 and EGL-9 prolyl hydroxylation is a mechanistic detail of this regulation and not the critical new finding. Although, we believe it does suggest where pathway analyses should be directed in the future. We also believe that our homeostatic feedback model for the regulation of HIF-1 (and cdo-1) by cysteine-derived H2S is new and exciting and provides insight into the logic of why HIF-1 might respond to H2S and promote the activity of cdo-1. Our work suggests that one reason for this intersection of hif-1 and cdo-1 is to sense and maintain cysteine homeostasis when cysteine is in excess.

Issue 9: the lack of biochemical approaches to probe the underlying mechanism of the prolyl hydroxylaseindependent activity of EGL-9.

Response 9: While not the primary focus of our current manuscript, we agree that this is an exciting area of future research. To uncover the prolyl hydroxylase-independent activity of EGL-9, we agree that a combination of approaches will be required including, biochemical, structure-function, and genetic.

Major Issues We Feel the Authors Should Address:Issue 10: One particularly glaring concern is that the authors really do not know the extent to which the prolyl hydroxylase activity is (or is not) impacted by the H487A mutation in egl-9(rae276). If there is a fair amount of enzymatic activity left in this mutant, then it complicates interpretation. The paper would be strengthened if the authors could show that the egl-9(rae276) eliminates most if not all prolyl hydroxylase activity. In addition, the authors may want to consider doing RNAi for egl-9 in the egl-9(rae276) mutant as a control, as this would support the claim that whatever non-hydroxylase activity EGL-9 may have is indeed the causative agent for the elevation of CDO-1::GFP. Without such experiments, readers are left with the nagging concern that this allele is simply a hypomorph for the single biochemical activity of EGL-9 (i.e., the prolyl hydroxylase activity) rather than the more interesting, hypothesized scenario that EGL-9 has multiple biochemical activities, only one of which is the prolyl hydroxylase activity.

Response 10: We have two lines of evidence that suggest the egl-9(rae276)-encoded H487A variant eliminates prolyl hydroxylase activity. First, Pan et al. 2007 (reference 57) demonstrate that when the equivalent histidine (H313) is mutated in human protein, that protein lacks detectible prolyl hydroxylase activity. Second, the phenotypic similarities caused by egl-9(rae276) and the vhl-1 null allele, ok161. Both alleles cause nearly identical activation of the Pcdo-1::GFP reporter transgene (Fig. 5C,D), and similarly impact the growth of the suox-1(gk738847) hypomorphic mutant (Table 1). This phenotypic overlap is highly relevant as the established role of VHL-1 is to recognize the hydroxyl mark conferred by the EGL-9 prolyl hydroxylase domain and promote the degradation of HIF-1. If EGL-9[H487A] had residual prolyl hydroxylase activity, we would expect the vhl-1(-) null mutant *C. elegans* to display more dramatic phenotypes than their egl-9(rae276) counterparts. This is not the case.

Issue 11: The authors observed that EGL-9 can inhibit HIF-1 and the expression of the HIF-1 target cdo-1 through a combination of activities that are (1) dependent on its prolyl hydroxylase activity (and subsequent VHL-1 activity that acts on the resulting hydroxylated prolines on HIF-1), and (2) independent of that activity. This is not a novel finding, as the authors themselves carefully note in their Discussion section, as this odd phenomenon has been observed for many HIF-1 target genes in multiple publications. While this manuscript adds to the description of this phenomenon, it does not really probe the underlying mechanism or shed light on how EGL-9 has these dual activities. This limits the overall impact and novelty of the paper.

Response 11: See response to Issues #8.

Issue 12: Cysteine dioxygenases like CDO-1 operate in an oxygen-dependent manner to generate sulfites from cysteine. CDO-1 activity is dependent upon availability of molecular oxygen; this is an unexpected characteristic of a HIF-1 target, as its very activation is dependent on low molecular oxygen. Authors neither address this in the text nor experimentally, and it seems a glaring omission.

Response 12: We agree this is an important point to raise within our manuscript. Although, despite its induction by HIF-1, there is no evidence that cdo-1 transcription is induced by hypoxia. In fact, in a genome wide transcriptomic study, cdo-1 was not found to be induced by hypoxia in *C. elegans* (Shen et al. 2005, reference 71).

We have newly commented on the use of molecular oxygen as a substrate by both EGL-9 and CDO-1 in our Discussion section. The mammalian oxygen-sensing prolyl hydroxylase (EGLN1) has been demonstrated to have high a Km value for O2 (high µM range). This likely allows EGLN1 to be poised to respond to small decreases in cellular oxygen from normal oxygen tensions. Clearly, CDO-1 also requires oxygen as a substrate, however the Km of CDO-1 for O2 is likely to be much lower, preventing sensitivity of the cysteine catabolism to physiological decreases in O2 availability. Although, to our knowledge, the CDO1 Km value for O2 has not been experimentally determined. We have added a new Discussion section where we address the conundrum about low oxygen inducing HIF-1 but oxygen being needed by CDO-1/CDO1.

Issue 13: The authors determined that the hypodermis is the site of the most prominent CDO-1::GFP expression, relevant to Figure 4. This claim would be strengthened if a negative control tissue, in the animal with the knockin allele, were shown. The hypodermal specific expression is a highlight of this paper, so it would make this article even stronger if they could further substantiate this claim.

Response 13: Our claim that the hypodermis is the critical site of cdo-1 function is based on; (i) our hands on experience looking at Pcdo-1::GFP, Pcdo-1::CDO-1::GFP, CDO-1::GFP (encoded by cdo-1(rae273)) and our reporting of these expression patterns in multiple figures throughout the manuscript and (ii) the functional rescue of cdo-1(-) phenotypes by a cdo-1 rescue construct expressed by a hypodermal-specific promoter (col10). We agree that providing negative control tissues would modestly improve the manuscript. However, we do not think that adding these controls will substantially alter the conclusions of the paper. Importantly, we acknowledge this limitation of our work with the sentence, “However, we cannot exclude the possibility that CDO-1 also acts in other cells and tissues as well.”

Minor issues to note:Issue 14: Mutants for hif-1 and cysl-1 are sensitive to exogenous cysteine levels, yet loss of CDO-1 expression is not sufficient to explain this phenomenon, suggesting other targets of HIF-1 are involved. Given the findings the authors (and others) have had showing a role for RHY-1 in sulfur amino acid metabolism, shouldn't the authors consider testing rhy-1 mutants for sensitivity to exogenous cysteine?

Response 14: To test the hypothesis that rhy-1(-) *C. elegans* might be sensitive to supplemental cysteine, we cultured wild type and rhy-1(-) animals on 0, 100, and 1000µM supplemental cysteine. At 0 and 100µM supplemental cysteine, neither wild-type nor rhy-1(-) animals display any lethality suggesting rhy-1 is not required for survival in the face of excess cysteine (Fig. 3D,E). We also cultured these same strains on 1000µM supplemental cysteine, a concentration that is highly toxic to wild-type animals (100% lethality). rhy1(-) animals were resistant to 1000µM supplemental cysteine with a substantial fraction of the population surviving overnight exposure to this lethal dose of cysteine. Similarly, egl-9(-) mutant *C. elegans* were also resistant to 1000µM supplemental cysteine. We propose that loss of egl-9 or rhy-1 activates HIF-1-mediated transcription which is priming these mutants to cope with the lethal dose of cysteine. These data are now presented in Figure 3D-F and presented in the Results section.

Issue 15: The cysteine exposure assay was performed by incubating nematodes overnight in liquid M9 media containing OP50 culture. The liquid culture approach adds two complications: (1) the worms are arguably starving or at least undernourished compared to animals grown on NGM plates, and (2) the worms are probably mildly hypoxic in the liquid cultures, which complicates the interpretation.

Response 15: We agree that it is possible that animals growing overnight in liquid culture are undernourished and mildly hypoxic. However, we are confident in our data interpretation as all our experiments are appropriately controlled. Meaning, control and experimental groups were all grown under the same liquid culture conditions. Thus, these animals would all experience the same stressors that come with liquid culture. Importantly, we never make comparisons between groups that were grown under different culture conditions (i.e. solid media vs. liquid culture).

Issue 16: An easily addressable concern is the wording of one of the main conclusions: that cdo-1 transcription is independent of the canonical prolyl hydroxylase function of EGL-9 and is instead dependent on one of EGL-9's non-canonical, non-characterized functions. There are several points in which the wording suggests that CDO-1 toxicity is independent of EGL-9. In their defense, the authors try to avoid this by saying,"EGL-9 PHD," to indicate that it is the prolyl hydroxylase function of EGL-9 that is not required for CDO-1 toxicity. However, this becomes confusing because much of the field uses PHD and EGL-9/EGLN as interchangeable protein names. The authors need to be clear about when they are describing the prolyl hydroxylase activity of EGL-9 rather than other (hypothesized) activities of EGL-9 that are independent of the prolyl hydroxylase activity.

Response 16: We appreciate the reviewer alerting us to this practice within the field. To avoid confusion, we have removed the “PHD” abbreviation from our manuscript and explicitly referred to the “prolyl hydroxylase domain” where relevant.

Issue 17: The authors state in the text, "the egl-9; suox-1 double mutants are extremely sick and slow growing." We appreciate that their "health" assay, based on the exhaustion of food from the plate, is qualitative. We also appreciate that it is a functional measure of many factors that contribute to how fast a population of worms can grow, reproduce, and consume that lawn of food. However, unless they do a lifespan assay and/or measure developmental timing and specifically determine that the double mutant animals themselves are developing and/or growing more slowly, we do not think it is appropriate to use the words "slow growing" to describe the population. As they point out, the rate of consumption of food on the plate in their health assay is determined by a multitude and indeed a confluence of factors; the growth rate is one specific one that is commonly measured and has an established meaning.

Response 17: We see how the phrase ‘slow growing’ might imply a phenotype that we have not actually assessed with this assay. Therefore, we have removed all claims about “slow growth” of the strains presented in Table 1 and have highlighted the assay more overtly in the results section. For example; “While egl-9(-) and suox-1(gk738847) single mutant animals are healthy under standard culture conditions, the egl-9(-); suox1(gk738847) double mutant animals are extremely sick and require significantly more days to exhaust their *E. coli* food source under standard culture conditions (Table 1).”

**Reviewer #1 (Recommendations For The Authors):**
Issue 18: Relevance could be addressed further in the text.

Response 18: We have added additional context for our work in the Discussion section. Please see our response to Issues #5, 6, 12, and 24.

Issue 19: Better appreciation and integration of the manuscript's findings with published studies would be appropriate.

Response 19: We have added additional context for our work in the Discussion section. Please see our response to Issues #5, 6, 12, and 24.

Issue 20: It might be perhaps relevant to test whether cdo-1 is relevant for hypoxia resistance since it appears to be a key target for hif-1.

Response 20: We agree that this is an interesting future direction, however given that cdo-1 mRNA is not induced by hypoxia (Shen et al. 2005) we have not prioritized these experiments for the current manuscript.

Issue 21: "egl-9 inhibits cdo-1 transcription in a prolyl-hydroxylase and VHL-1-independent manner" should be tempered. vhl-1 mutants and egl-9 hydroxylase point mutant still have significant induction of the reporter.

Response 21: Thank you for identifying this oversight. We have modified the Figure 5 legend title to read, “egl9 inhibits cdo-1 transcription in a largely prolyl-hydroxylase and VHL-1-independent manner.”

Issue 22: Please use line numbers in the future for easier tracking of comments.

Response 22: We shall.

Issue 23: Abstract and elsewhere, "high cysteine activates...", should be rephrased to "high levels of cysteine".

Response 23: We have made this change throughout the manuscript.

**Reviewer #3 (Recommendations For The Authors):**
Issue 24: The authors discuss CDO1 in the context of tumorigenesis, as well as the potential regulation between cysteine and the hypoxia response pathway. Thus, I was surprised that there was no mention of the foundational Bill Kaelin paper (Briggs et al 2016) showing how the accumulation of cysteine is related to tumorigenesis, and that cysteine is a direct activator of EglN1. Puzzling that CDO1 is a tumor suppressor: you lose it, cysteine can accumulate and activate EglN1, causing HIF1 turnover. How do the authors reconcile their results with this paper? I was also surprised that there was no mention in the Discussion of the role of hydrogen sulfide, cysteine metabolism, and CTH and CBS in oxygen sensation in the carotid body given the role they play there. Seems important to discuss this issue.

Response 24: We have added new sections to our Discussion that consider the relationship between our work and Briggs et al. 2016 as well as mentioned the role of CTH and H2S in the mammalian carotid body.

Issue 25: The abstract has a variety of contradictory statements. For example, the authors state that "HIF-1mediated induction of cdo-1 functions largely independent of EGL-9," but then go on to conclude in the final sentence that cysteine stimulates H2S production, which then activates EGL-9 signaling, which then increases HIF-1-mediated transcription of cdo-1. A quick reading of the abstract leaves the reader uncertain whether EGL-9 is or is not involved in this regulation of cdo-1 expression. In addition, the conclusion sentence implies that activation of the EGL-9 pathway increases HIF-1-mediated transcription, yet it is well established that EGL-9 is an inhibitor of HIF-1. The abstract fails to deliver a clear summary of the paper's conclusions. Perhaps consider this alternative (changes in capital letters):The amino acid cysteine is critical for many aspects of life, yet excess cysteine is toxic. Therefore, animals require pathways to maintain cysteine homeostasis. In mammals, high cysteine activates cysteine dioxygenase, a key enzyme in cysteine catabolism. The mechanism by which cysteine dioxygenase is regulated remains largely unknown. We discovered that *C. elegans* cysteine dioxygenase (cdo-1) is transcriptionally activated by high cysteine and the hypoxia inducible transcription factor (hif-1). hif-1- dependent activation of cdo-1 occurs downstream of an H2S-sensing pathway that includes rhy-1, cysl-1, and egl-9. cdo-1 transcription is primarily activated in the hypodermis where it is sufficient to drive sulfur amino acid metabolism. EGL-9 and HIF-1 are core members of the cellular hypoxia response. However, we demonstrate that the mechanism of HIF-1-mediated induction of cdo-1 IS largely independent of EGL-9 prolyl hydroxylASE ACTIVITY and the von Hippel-Lindau E3 ubiquitin ligase. We propose that the REGULATION OF cdo-1 BY HIF-1 reveals a negative feedback loop for maintaining cysteine homeostasis. High cysteine stimulates the production of an H2S signal. H2S then ACTS THROUGH the rhy-1/cysl-1/egl-9 signaling pathway DISTINCTLY FROM THEIR ROLE IN HYPOXIA RESPONSE TO INCREASE HIF-1-mediated transcription of cdo-1, promoting degradation of cysteine via CDO-1.

Response 25: We agree that the abstract could be clearer. We believe this concern stems from the fact that we did not discuss our initial screen in the abstract. Thus, we failed to establish a role for egl-9 in the regulation of cdo-1. To remedy this, we have modified the abstract as suggested by the reviewer and added additional context. We believe that these changes improve the clarity of the Abstract substantially.

Issue 26: An easily addressable concern involves the "dark" microscopy controls showing lack of fluorescence from a nematode. In these dark negative control micrographs, the authors should draw dotted outlines around where the worms are or include a brightfield image next to the fluorescence image. On a computer screen, it is in fact possible to make out the worms. Yet, when printed out, the reader must assume there are worms in the dark images. Additionally, we realize that adjusting fluorescence so that wild-type CDO-1 expression can be seen will result in oversaturation of the egl-9 and rhy-1; cdo-1 doubles; however, this would be a useful figure to add into the supplement to both provide a normal reference of CDO-1 low-level expression and a demonstration of just how bright it is in the mutant backgrounds. It would also be useful for you to please report your exposure settings for purposes of reproducibility.

Response 26: As suggested, we have added dotted lines around the location of the *C. elegans* animals in all images where GFP expression is low or basal. We have also reported the exposure times for each image in the appropriate figure legends.

Issue 27: This title is quite generic and doesn't even mention the main players (CDO-1 and sulfite metabolism).

Response 27: We have updated our title to call attention to cysteine dioxygenase. The improved title is: “Hypoxia-inducible factor induces cysteine dioxygenase and promotes cysteine homeostasis in *Caenorhabditis elegans*”

Issue 28: The authors mention two disorders in which CDO-1 plays a pathogenic role: MoCD and ISOD. We recommend switching the order in which the authors mention these, as the remainder of the paragraph is about MoCD. Also, they should write out the number "2" in the first sentence of that paragraph.

Response 28: We have made the suggested changes.

Issue 29: The authors state in the main text, "...to ubiquitinate HIF-1, targeting it for degradation by the proteosome." Here, they should refer to the pathway in Figure 5a.

Response 29: We have made the suggested change.

Issue 30: The authors state in the main text, "Elements of the HIF-1 pathway have emerged..." which is vague and confusingly worded. Change to, "Members of the HIF-1 pathway and its targets have emerged from *C. elegans* genetic studies."

Response 30: We have made the suggested change.

Issue 31: Clarify in the figure legends that supplemental cysteine did not affect the mortality of worms that were imaged.

Response 31: We have added this note to Figure 3A and Figure S3A.

Issue 32: Figure 1b. "the cdo-1 promoter is shown..." Add: "as a straight line" to the end of this phrase.

Response 32: We have made the suggested change.

Issue 33: The authors should consider changing the red text in Figure 1 to magenta, which tends to be more readable for people who have limited color vision.

Response 33: We have adjusted the colors in Figure 1 as suggested.

Issue 34: Figure 2, legend title. Consider changing "hif-1" to "HIF-1," as well as rhy-1, cysl-1, and egl-9. In this case, they are talking about proteins, not mutants or genes. This will make the paper easier to follow for readers who lack a *C. elegans* background.

Response 34: We have made the suggested change.

Issue 35: Figure 5, caption text. "...indicates weak similarity." Add, "amongst species compared."

Response 35: We have made the suggested change.

Issue 36: It is starting to become a standard for showing the datapoints in bar graphs. Although this is done in many graphs in the paper, it should also be done for Figure S1 and Figure 4C.

Response 36: We have made the suggested change.

Issue 37: An extensive ChIP-seq and RNA-seq analysis of *C. elegans* HIF-1 was recently published (Vora et al, 2022), which the authors should reference in support of the regulation of CDO-1 transcription by HIF-1 in their description of published expression studies of the pathway (Results section, page 4). Indeed, Vora et al were key generators of the ChIP-seq data cited in Warnhoff et al but not included as authors in theModERN/ModENCODE publication: their contributions were published separately in Vora et al and should be acknowledged equivalently.

Response 37: We appreciate the reviewer pointing this detail out and we have added the correct citation as indicated.